# HawkI: Homography & Mutual Information Guidance for 3D-free Single Image to Aerial View

## Abstract

We present HawkI, for synthesizing aerial-view images from text and an exemplar image, without any additional multi-view or 3D information for finetuning or at inference. HawkI uses techniques from classical computer vision and information theory. It seamlessly blends the visual features from the input image within a pretrained text-to-2D-image stable diffusion model with a test-time optimization process for a careful bias-variance (fidelity-diversity/viewpoint) trade-off, which uses an *Inverse Perspective Mapping (IPM) homography transformation* to provide subtle cues for aerial-view synthesis. At inference, HawkI employs a unique *mutual information guidance* formulation to steer the generated image towards faithfully replicating the semantic details of the input-image, while maintaining a realistic aerial perspective. Mutual information guidance maximizes the semantic consistency between the generated image and the input image, without enforcing pixel-level correspondence between vastly different viewpoints. Through extensive qualitative and quantitative comparisons against text + exemplar-image based methods and 3D/ multi-view based novel-view synthesis methods on proposed synthetic and real datasets, we demonstrate that our method achieves a significantly better bias-variance trade-off towards generating high fidelity aerial-view images. [1]

## 1 Introduction

Widely available text-to-image models such as Stable Diffusion (Rombach et al., 2022), trained on large-scale text and 2D image data, contain rich knowledge of the 3D world. They are capable of generating scenes from various viewpoints, including aerial views. However, due to limitations of the expressiveness of the text, we may not be able to completely describe the precise scene that we wish to generate. Moreover, the generative capabilities of the pretrained model is constrained by the aerial-view images in the dataset that it was trained on, which is typically limited. Consequentially, along with text, it is beneficial to use an easily available representative front-view image describing the aerial view of the scene we wish to generate. The task of generating aerial-view images from a given input image and its text description finds applications in the generation of realistic diverse aerial view synthetic data for improved aerial view perception tasks (Kothandaraman et al., 2022; Li et al., 2021; Kothandaraman et al., 2023a; Choi et al., 2020; Barekatain et al., 2017), and weak supervision for cross-view synthesis applications (Ma et al., 2022) such as localization and mapping (Hu et al., 2018), autonomous driving (Chen et al., 2017), augmented and virtual reality (Emmaneel et al., 2023), 3D reconstruction (Wang et al., 2021), medical imaging (van Tulder et al., 2021), drone-enabled surveillance (Ardeshir & Borji, 2018).

Aerial-view images corresponding to text and an input image can be sampled using text-to-3D and novel view synthesis (NVS) (Liu et al., 2023b; Poole et al., 2022). These methods sample different camera viewpoints by explicitly specifying the camera angle. However, they often need to trained on enormous, large-scale datasets with 3D details and scenes from multiple views. Is it possible for text-to-image(2D) diffusion models to generate aerial-view images without any multi-view or 3D information?

Another closely related task is image editing (Kawar et al., 2023) and personalization (Ruiz et al., 2023a), where the goal is to use an input image and a target text to generate an image consistent with both inputs. These methods are generally successful in performing a wide range of non-rigid transformations including text-controlled view synthesis. However, the large translation required for aerial view synthesis makes them

---

[1]All code and data will be public.

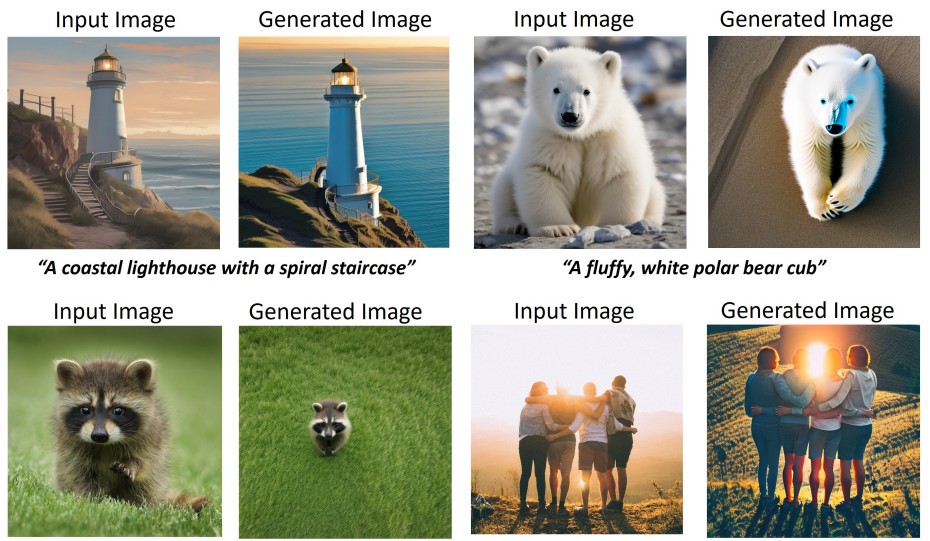

Figure 1: HawkI generates aerial-view images from a text description and an exemplar input image. It builds on a text to 2D image stable diffusion model and does not require any additional 3D or multi-view information at fine-tuning or inference.

sub-optimal (Poole et al., 2022). This is due to bias-variance trade-off issues, even more amplified when only a single input image is provided. Aerial Diffusion (Kothandaraman et al., 2023b) attempted to alleviate this bias-variance trade-off, but at the cost of per-sample hyperparameter tuning, residual diagonal artifacts in many of the generated images arising from direct finetuning on sub-optimal homography projections and severe performance drops on complex scenes with multiple objects.

**Main contributions.** We propose HawkI for aerial view synthesis, guided by text and a single input image. Our method leverages text-to-image diffusion models for prior knowledge and does not require any 3D or multi-view data. Since explicitly specifying camera details in text descriptions isn't always possible, similar to prior work on text-based viewpoint generation (Ruiz et al., 2023a; Kothandaraman et al., 2023b), we consider any generated image with a significantly higher viewpoint and altitude compared to the original image to be an aerial view. HawkI fuses techniques from classical computer vision and information theory within a stable diffusion backbone model to guide the synthesis of the aerial-view image. The key novel components of our algorithm include:

1. **Test-time optimization**: This step enables the model to acquire the characteristics of the input image, while maintaining sufficient variability in the embedding space for aerial-view synthesis. We condition the embedding space by sequentially optimizing the CLIP text embedding and the LoRA layers corresponding to the diffusion UNet on the input image and its Inverse Perspective Mapping (IPM) homography transformation in close vicinity. In addition to creating variance, IPM provides implicit guidance towards the direction of transformation for aerial-view synthesis.

2. **Mutual Information Guided Inference**: This step generates a semantically consistent aerial-view image while accounting for viewpoint differences. Unlike conventional approaches (Bansal et al., 2023; Epstein et al., 2024) that rely on restrictive pixel-level constraints (often ineffective for different viewpoints), we propose a mutual information vastly guidance formulation. Mutual information guidance, rooted in information theory, ensures consistency between the contents of the generated image and the input image by maximizing the information contained between the probability distributions of the input image and the generated aerial image.

Our method performs *inference-time* optimization on the given text-image inputs and does not require a dataset to train on, hence, it is easily applicable to any in-the-wild image. To test our method, we collect a

diverse set of synthetic images (from Stable Diffusion XL) and real images (from Unsplash), spanning across natural scenes, indoor scenes, human actions and animations. Qualitative and quantitative comparisons with prior work, on metrics such as CLIP (Radford et al., 2021) (measuring viewpoint and text consistency) and SSCD (Pizzi et al., 2022), DINOv2 (Oquab et al., 2023) (measuring consistency w.r.t. input image), demonstrate that HawkI generates aerial-view images with a significantly better fidelity-diversity/ viewpoint (or bias-variance) trade-off. We also present extensive ablation experiments and comparisons with 3D-based novel-view synthesis methods highlighting the benefits of our 3D-free classical guidance approaches. Our method can also be extended to generate more views that can be text-controlled (such as 'side view', bottom view', 'back view'), as evidenced by our results.

## 2 Related work

**3D and novel view synthesis.** Novel view synthesis (Wiles et al., 2020; Tucker & Snavely, 2020; Park et al., 2017) from a single image is an active area of research in generative AI. Many methods (Tancik et al., 2022; Jain et al., 2021; Gu et al., 2023; Zhou & Tulsiani, 2023) use NeRF based techniques. Nerdi (Deng et al., 2023) use language guidance with NeRFs for view synthesis. Many recent methods use diffusion (Liu et al., 2023a; Shi et al., 2023c; Liu et al., 2023b; Qian et al., 2023; Shi et al., 2023b;a; Burgess et al., 2023; Sargent et al., 2023) to sample different views. 3D generation methods (Poole et al., 2022; Lin et al., 2023; Xu et al., 2023; Raj et al., 2023; Chen et al., 2023) use text to guide the reconstruction. All of these methods use large amounts of multi-view and 3D data for supervised training. Methods like Zero-1-to-3 (Liu et al., 2023b) and Zero-123++ (Shi et al., 2023a) use a pretrained stable diffusion (Rombach et al., 2022) model, along with large data for supervised training, to learn different camera viewpoints. 3D-free methods such as Free3D (Zheng & Vedaldi, 2023) still require multi-view and 3D information while training.

**Warping, scene extrapolation and homography.** Scenescape (Fridman et al., 2024), DiffDreamer (Cai et al., 2023) and similar methods (Wiles et al., 2020; Rockwell et al., 2021; Chen & Koltun, 2017) estimate a depth map, reproject the pixels into the desired camera perspective and outpaint the scene. Again, these methods require 3D and multi-view information at training stage. Using a homography to estimate the scene from an aerial perspective is highly inaccurate, hence, attempting to create realistic aerial view images by simply filling in missing information based on the homography (outpainting) leads to poor outcomes. Homography maps have also been used in various deep learning based computer vision solutions (Zhang et al., 2020; Ding & Tao, 2017; Liu & Li, 2023; Gu et al., 2022; D'Amicantonio et al., 2024).

**Image manipulation/ personalization.** Diffusion models have emerged as successful tools for single image editing and personalization. Methods such as DreamBooth (Ruiz et al., 2023a), DreamBooth LoRA (Hu et al., 2021), HyperDreamBooth (Ruiz et al., 2023b), Textual Inversion (Gal et al., 2022), Custom Diffusion (Kumari et al., 2023) are able to generated personalized images of subjects. Image editing and manipulation methods such as Imagic (Kawar et al., 2023), Paint-by-Example (Yang et al., 2023), ControlNet (Zhang et al., 2023), DiffEdit (Couairon et al., 2022), steerability (Jahanian et al., 2019), visual anagrams (Geng et al., 2024) are able to edit images to perform non-rigid transformations and also use exemplar signals for guidance. However, these methods can either generate aerial images with low fidelity w.r.t. the input image or generate high-fidelity images with viewpoints very close to the input image.

**Cross-view synthesis.** Prior work on cross-view synthesis (Regmi & Borji, 2019; Tang et al., 2019; Ren et al., 2022; Toker et al., 2021; Ding et al., 2020; Ma et al., 2022; Liu et al., 2021; Shi et al., 2022; Liu et al., 2020; Ren et al., 2021; Liu et al., 2022; Wu et al., 2022; Shen et al., 2021; Ammar Abbas & Zisserman, 2019; Zhao et al., 2022) are data intensive - they use paired data and modalities such as semantic maps, depth, multi-views, etc within their architectures. Aerial Diffusion (Kothandaraman et al., 2023b) uses text and an exemplar image for the task by alternating sampling between viewpoint and homography projects. However, the generated images have diagonal artifacts with poor quality results for complex scenes that typically contain more than one object and requires manual per-sample hyperparameter tuning.

**Guidance techniques in diffusion.** Guidance methods (Ho & Salimans, 2022; Bansal et al., 2023; Dhariwal & Nichol, 2021; Nair et al., 2023) have been used to control and guide diffusion denoising towards semantic maps, image classes, etc. These guidance techniques cannot enforce view-invariant image-image similarity, critical for aligning the contents in two images with vastly different viewpoints.

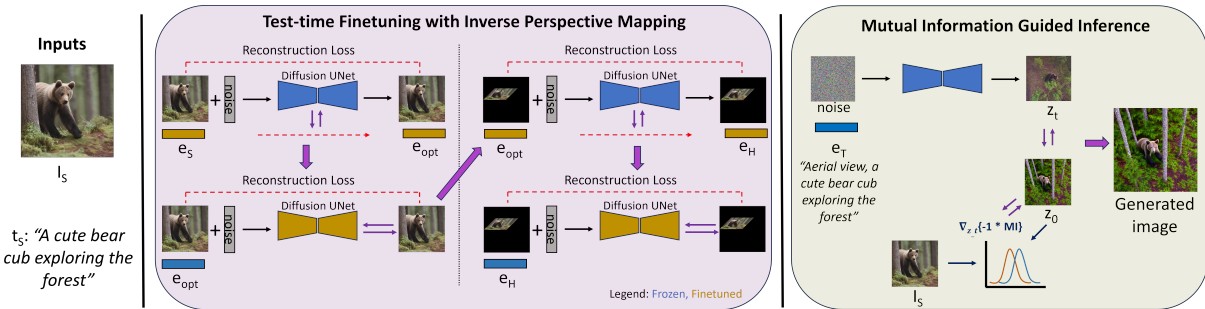

Figure 2: **Overview.** HawkI generates aerial-view images, using a text description and a single image $I_S$ as supervisory signals. It builds on a pretrained text-to-image diffusion model, and does not use any 3D or multi-view information. It performs test-time finetuning to optimize the text embedding and the diffusion model to reconstruct the input image and its inverse perspective mapping in close vicinity. Such a mechanism enables the incorporation of image specific knowledge within the model, while retaining its imaginative capabilities (or variance). At inference, HawkI uses mutual information guidance to maximize the information between the probability distributions of the generated image and $I_S$, to generate a high-fidelity aerial-view image.

## 3 Method

We present HawkI to generate aerial view images using a single input image $I_S$ and its text description $t_S$ (e.g. 'a cosy living room', can be obtained using the BLIP-2 model (Li et al., 2023)). We do not use any training data or 3D/multi-view data. We leverage the pretrained text-to-2D image stable diffusion (Rombach et al., 2022) model to serve as a strong prior, and utilize classical computer vision and information theory principles to achieve the desired goal in a holistic manner. We present an overview of our method in Figure 2.

- *Test-time optimization*: We perform multi-step test-time optimization to incorporate the input image $I_S$ within the pretrained model, at an appropriate fidelity-diversity trade-off. Specifically, we optimize the CLIP text-image embedding and the LoRA layers in the diffusion UNet sequentially on the input image and its inverse perspective mapping, in close vicinity. This additionally conditions the embedding space viewpoint transformations, along with acquiring the characteristics of the input image.

- *Inference*: To generate the aerial-view image, we use the target text description $t_T$, of the form **'aerial view, '** $+ t_S$ (e.g. 'aerial view, a cosy living room'). To ensure that the generated aerial image is semantically close to the input image, we use mutual information guidance.

Next, we describe our method in detail.

### 3.1 Test-time optimization

The text-to-2D image stable diffusion model has knowledge of the 3D world as a consequence of the large amount of diverse data it has been trained on. It understands (Schuhmann et al., 2022) different viewpoints, different styles, backgrounds, etc. Image editing and personalization methods such as DreamBooth (Ruiz et al., 2023a), DreamBooth LoRA (Hu et al., 2021), Imagic (Kawar et al., 2023), SVDiff (Han et al., 2023) exploit this property to perform transformations such as making a standing dog sit and generating it in front of the Eiffel tower. At a high level, the standard procedure adopted by these methods to generate edited or personalized images is to finetune the model on the input image, followed by inferencing. These methods are however not very successful in text-guided aerial view synthesis, which demands a large transformation. Specifically, directly finetuning the diffusion model on $e_S$ to reconstruct $I_S$ results in severe overfitting, where $e_S$ is the CLIP text embedding for $t_S$. This makes it difficult for the model to generate large variations to the scene required for aerial view synthesis.
We propose a four-step finetuning approach to enable the model to learn the characteristics of $I_S$, while ensuring sufficient variance for aerial view generation.

### 3.1.1 Optimization using $I_S$:

In the first step (Kawar et al., 2023), we start from $e_S$ and compute the optimized CLIP text embedding $e_{opt}$ to reconstruct $I_S$ using a frozen diffusion model UNet using the denoising diffusion loss function $L$ (Ho

et al., 2020).

$$\min_{e_{opt}} \sum_{t=T}^{0} L(f(x_t, t, e_{opt}; \theta), I_S), \tag{1}$$

where $t$ is the diffusion timestep and $x_t$ is the latents at time $t$. This formulation allows us to find the text embedding that characterizes $I_S$ better than the generic text embedding $e_S$.

Next, to enable $e_{opt}$ accurately reconstruct $I_S$, we optimize the diffusion UNet using the denoising diffusion objective function. Note that we insert LoRA layers within the attention modules in the diffusion UNet and finetune only the LoRA layers with parameters $\theta_{LoRA}$, the rest of the UNet parameters are frozen,

$$\min_{\theta_{LoRA}} \sum_{t=T}^{0} L(f(x_t, t, e_{opt}; \theta), I_S). \tag{2}$$

While optimizing $e_{opt}$ instead of $e_S$ to reconstruct $I_S$ ensures lesser bias (or more variance), the embedding space is still not sufficiently conditioned to generate an aerial view of the image.

### 3.1.2 Optimization using inverse perspective mapping.

Inverse perspective mapping (IPM) (Szeliski, 2022) is a homography transformation from classical computer vision to generate the aerial-view of an image from its ground-view. Despite not being accurate, it can provide pseudo weak supervision for the generation of the aerial image and also add more variance to the embedding space. We denote the inverse perspective mapping of the input image by $I_H$, computed following (Kothandaraman et al., 2023b). We perform the following optimization steps to condition the embedding space towards the desired viewpoint transformation. To find the text embedding $e_H$ that best characterises $I_H$, we start from $e_{opt}$ and optimize the text embedding with a frozen diffusion model, similar to Equation 1. Finding $e_H$ in the vicinity of $e_{opt}$ instead of $e_S$ ensures that the text-image space corresponding to $e_S$ doesn't get distorted to generate the poor quality IPM image. Next, we finetune the diffusion model using the denoising diffusion objective function to reconstruct $I_H$ at $e_H$, similar to Equation 2. Again, only the LoRA layers are finetuned, the rest of the UNet is frozen.

Note that we find $e_{opt}$ and $e_H$ by optimizing $e_S$ and $e_{opt}$, respectively for a small number of iterations. We need to ensure that $e_S$, $e_{opt}$ and $e_H$ are all in close vicinity. Our finetuning approach conditions the embedding space to encapsulate the details of $I_S$ and viewpoint, while having sufficient variance to generate large transformations required for the generation of the aerial image.

### 3.2 Mutual Information Guided Inference

Our next step is to use the finetuned diffusion model to generate the aerial view image for the text prompt $t_T$. The text embedding for $t_T$ is $e_T$. Diffusion denoising, conditioned on $e_T$ is capable of generating aerial images corresponding to $I_S$. However, oftentimes, the contents of the generated aerial view image does not align well with the contents of $I_S$. Consequently, to ensure high fidelity generations, our goal is to guide the contents of the aerial view image towards the contents of $I_S$.

Similarity measures such as L1 distance, cosine similarity are capable of providing this guidance. However, they are not invariant to viewpoint/ structure. Since we want the two images to be similar (while observed from different viewpoints), using metrics that impose matching at the pixel (or feature) level is not the best approach. Rather, it is judicious to use the probability distribution of the features.

In information theory, mutual information quantifies the 'amount of information' obtained about one random variable by observing the other random variable. Mutual information has been used (Viola & Wells III, 1997; Maes et al., 1997; Klein et al., 2007; Xian et al., 2023) to measure the similarity between images in various computer vision tasks such as medical image registration, frame sampling, etc. It yields smooth cost functions for optimization (Thomas, 1991). The mutual information between two probability distribution functions (pdf) $p(x), p(y)$ for two random variables $\mathcal{X}, \mathcal{Y}$ is defined as $I(\mathcal{X}, \mathcal{Y}) = H(\mathcal{X}) + H(\mathcal{Y}) - H(\mathcal{X}, \mathcal{Y})$ where $H(\mathcal{X}), H(\mathcal{Y})$ are the entropies of $p(x), p(y)$ and $H(\mathcal{X}, \mathcal{Y})$ is the joint entropy. Entropy of a random variable $X$ is a measure of its uncertainity, $H(\mathcal{X}) = -\sum_{x \in \mathcal{X}} p_X(x) log(p_X(x))$;

$$H(\mathcal{X}, \mathcal{Y}) = -\sum_{(x,y) \in \mathcal{X}, \mathcal{Y}} p_{XY}(x, y) log(p_{XY}(x, y)).$$

Thus,

$$I(\mathcal{X}, \mathcal{Y}) = - \sum_{(x,y) \in \mathcal{X}, \mathcal{Y}} p_{XY}(x,y) \frac{log(p_{XY}(x,y))}{p_X(x)p_Y(y)}.$$

Hence, mutual information, in some sense, measures the distance between the actual joint distribution between two probability distributions and the distribution under an assumption that the two variables are completely independent. Thus, it is a measure of dependence (Hyvärinen & Oja, 2000) and can be used to measure the information between two images. In order to maximize the similarity in content between $I_S$ and the generated aerial image, we maximize the mutual information between them. We define our mutual information guidance function as follows.

Let $z_t$ denote the predicted latents at timestep $t$. We denote $z_{0,t}$ as the latents of the final predicted image extrapolated from $z_t$ i.e. if the denoising were to proceed in a vanilla fashion in the same direction that computed $z_t$, the latents of the final predicted image would be $z_{0,t}$. At every step of sampling (except the final step), we wish to maximize the mutual information between $z_{0,t}$ and $z_S$ where $z_S$ are the latents corresponding to $I_S$. Hence, the guidance function we wish to maximize is, $G_{MI} = I(z_{0,t}, z_S)$.

The computation of mutual information requires us to compute the marginal and joint probability density functions (pdf) of $z_{0,t}$ and $z_S$. We construct 2D histograms of the latents (by reshaping the latents of size $C \times H \times W$ into $C \times HW$) and compute their marginal pdfs. The joint pdfs can then be computed from the marginal pdfs, which can be plugged into the formula for mutual information. Next, we will use $G_{MI}$ to guide the generation of the aerial image.

Guidance techniques such as classifier-free guidance (Ho & Salimans, 2022), universal guidance (Bansal et al., 2023) and steered diffusion (Nair et al., 2023) modify the sampling method to guide the image generation with feedback from the guidance function. The gradient of the guidance function w.r.t. the predicted noise at timestep $t$ is an indicator of the additional noise that needs to be removed from the latents to steer the generated image towards the guidance signal. Synonymously, the gradient of the guidance function w.r.t. the predicted latents is an indicator of the direction in which the latents need to move in order to maximize their alignment with the guidance function. Specifically, at every step of sampling (except the final step), we modify the predicted latents $z_t$ as $\hat{z}_t = z_t - \lambda_{MI} \nabla_{z_t}(-1 * G_{MI})$. Note that we use the negative of the mutual information to compute the gradients since we want to maximize the mutual information between the generated latents and the input image.

## 4 Experiments and Results

**Data.** We collect a synthetic dataset, HawkI-Syn and a real dataset, HawkI-Real. Both datasets contain images across a wide variety of categories including indoor scenes, natural scenes, human actions, birds/ animals, animations, traffic scenes and architectures. HawkI-Syn contains 500 images that were generated using Stable Diffusion XL (Podell et al., 2023). To generate the text prompts for the generated images in HawkI-Syn, we used Large Languuage Models (LLMs) such as ChatGPT and Bard. HawkI-Real contains 139 images downloaded from the Unsplash website, the text descriptions for these images were obtained using the BLIP-2 (Li et al., 2023) model.

**Training details.** We use the stable diffusion 2.1 model in all our experiments, ablations and comparisons. All our images (except for images in HawkI-Real) are at a resolution of $512 \times 512$. With respect to $I_S$, we train the text embedding and the diffusion model for $1,000$ and $500$ iterations respectively, at a learning rate of $1e-3$ and $2e-4$ respectively. Using 1000 iterations to optimize the text embedding ensures that the text embedding $e_{opt}$ at which $I_S$ is reconstructed is not too close to $e_S$, which would make it biased towards $I_S$ otherwise. Similarly, it is not too far from $e_S$ either, hence the text embedding space learns the characteristics of $I_S$. With respect to $I_H$, we train the text embedding and the diffusion UNet for 500 and 250 iterations respectively. We want $e_H$ to be in close vicinity of $e_{opt}$; we train the diffusion model for just 250 iterations so that the model does not completely overfit to $I_H$. The role of $I_H$ is to create variance and provide pseudo supervision, it is not an accurate approximation of the aerial view. We set the hyperparameter for mutual information guidance at $1e-5$ or $1e-6$, the inference is run for 50 steps.

**Computational cost.** For each input image, HawkI takes 3.5 minutes to perform test-time optimization on one NVIDIA A5000 GPU with 24 GB memory. The inference time is consistent with that of Stable Diffusion, about 7 seconds to generate each sample with 50 denoising steps. The computational cost is on

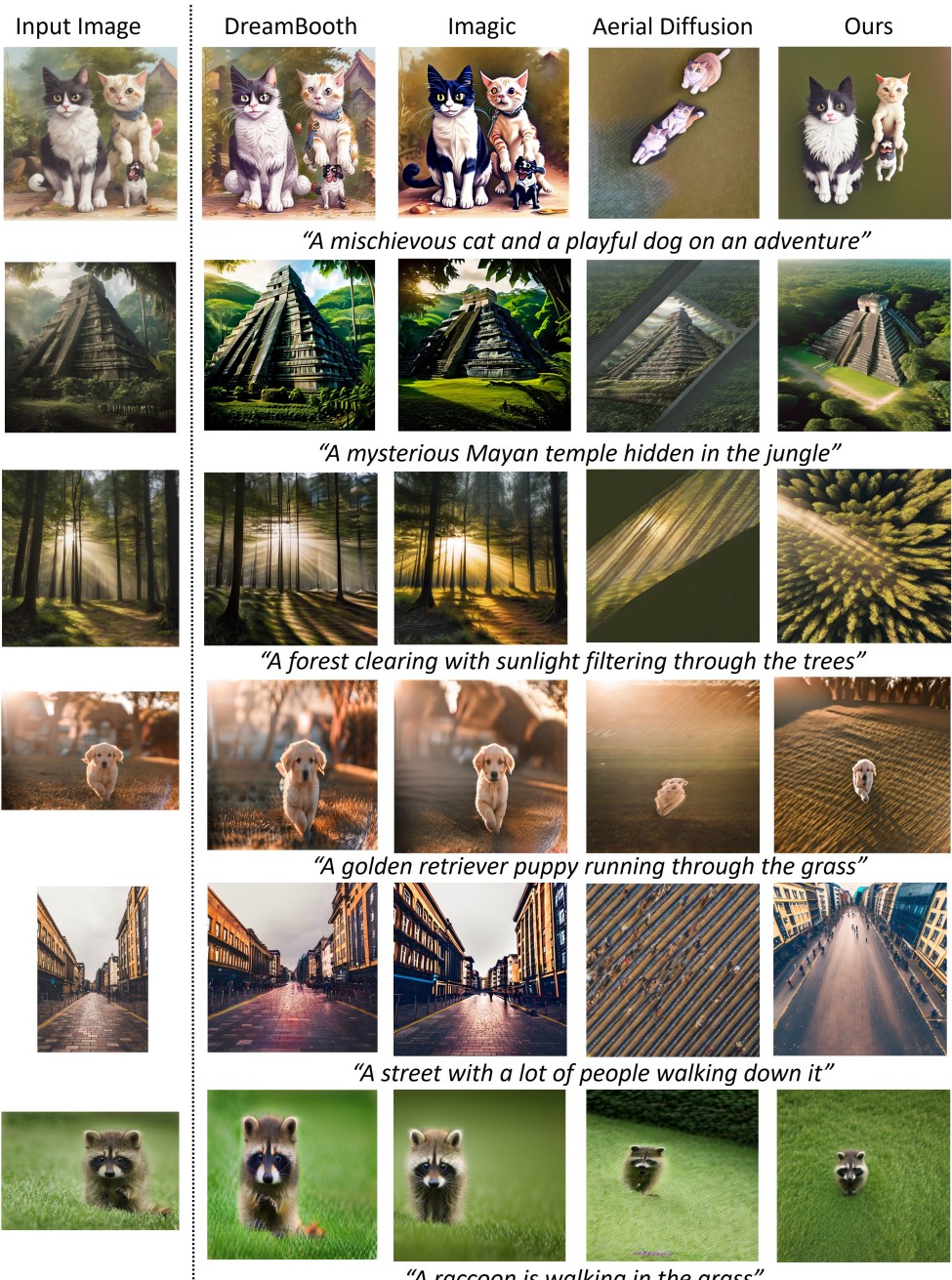

Figure 3: Compared to state-of-the-art text + exemplar image based methods, HawkI is able to generate images that are "more aerial", while being consistent with the input image. The top three images are from the HawkI-Syn dataset, the bottom three images are from the HawkI-Real dataset.

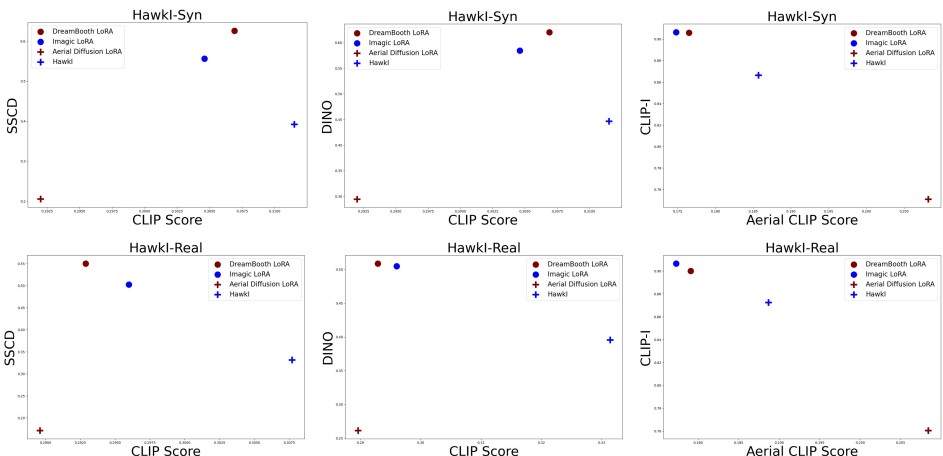

Figure 4: HawkI achieves the best viewpoint-fidelity trade-off amongst prior work on text + exemplar image based aerial-view synthesis, on various quantitative metrics indicate of text-alignment (for viewpoint and a broad description of the scene) and image alignment (for fidelity w.r.t. input image).

par with text-based image personalization (Ruiz et al., 2023a; Kawar et al., 2023) and text-based aerial-view synthesis (Kothandaraman et al., 2023b) methods. The number of network parameters is also consistent across all these models, we use the Stable Diffusion v2.1 + LoRA backbone across methods.

**Quantitative evaluation metrics.** We follow prior work on text-based image editing/ personalization and text-based view synthesis to evaluate our method:

- Viewpoint and text alignment: We use the text description 'aerial view, ' + $t_S$ and 'aerial view', along with the generated image, to compute the CLIP-Score (Radford et al., 2021) and the A-CLIP Score respectively. The former indicates alignment of the generated image with the detailed textual description of the image describing the contents along with the viewpoint; the latter focuses more on the viewpoint.
- Image fidelity and 3D coherence: To evaluate the overall alignment of the contents of the generated aerial-view image with the input image, we compute the CLIP-I score (Ruiz et al., 2023a) which measures the cosine similarity between the embeddings of the aerial-view image and the input image in the CLIP space. For a better indicator of the fidelity and 3D coherence between the two images, we also use the self-supervised similarity detection metrics, DINOv2 (Caron et al., 2021; Oquab et al., 2023) and SSCD (Pizzi et al., 2022).
- Top-1 accuracy on a downstream UAV task.
- Ground-truth comparison with images sampled from 3D models using CLIP, LPIPS, and DINO metrics.

Higher values are desired for each of these metrics. Viewpoint faithfulness and fidelity w.r.t input image are a direct result of the fidelity-diversity trade-off of the model, and high values for both are desired. However, as noted by Blau et. al. (Blau & Michaeli, 2018), maximizing both is not straightforward; inevitably one of the factors will degrade in response to the improvement in the other. For each input image, we generate 5 aerial images, with random noise initializations, and choose the image with the highest CLIP + SSCD score (since CLIP is an indicator of viewpoint + content alignment and SSCD score measures the fidelity w.r.t. the input image).

### 4.1 Comparisons against text + exemplar image based methods

We compare our method with DreamBooth LoRA (Ruiz et al., 2023a), a text-based image personalization method; Imagic LoRA (Kawar et al., 2023), a text-based image editing method; and Aerial Diffusion LoRA (Kothandaraman et al., 2023b), a method for text-based aerial image generation from a single image. We keep the backbone stable diffusion model, image prompts, training details, and evaluation method consistent across all comparisons.

We show qualitative results in Figure 3. Our method is able to generate aerial views as per input image guidance across a diverse set of scenes. Our method generates results that are more aerial in viewpoint

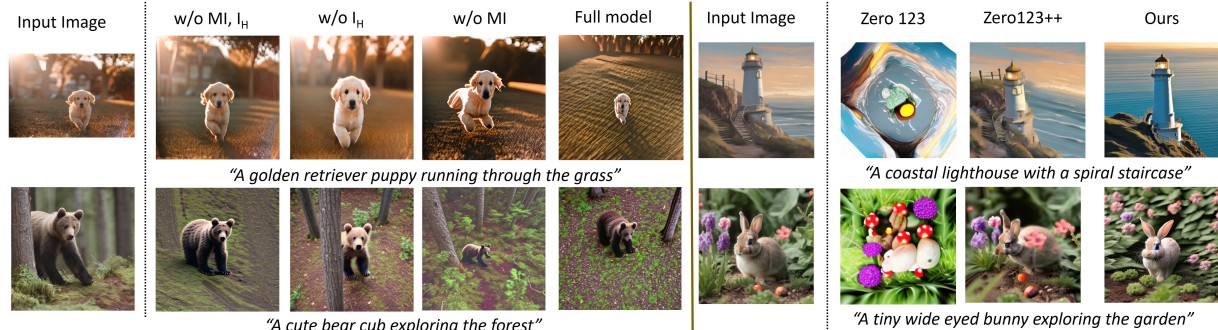

Figure 5: **(Left figure.)** Ablation experiments show that Inverse Perspective Mapping helps in the generation of images that are aerial, mutual information guidance helps in preserving the contents w.r.t. input image. **(Right figure.)** We compare with latest related work on novel view synthesis: Zero-1-to-3(Liu et al., 2023b) and Zero123++ (Shi et al., 2023a). Both methods use the pretrained text-to-2D-image stable diffusion model along with the 800k 3D objects dataset, Objaverse (Deitke et al., 2023), for training. Our method uses just the pretrained text-to-2D-image stable diffusion model to generate better results for the task of aerial view synthesis, guided by text and a single input image.

than DreamBooth and Imagic, while being largely consistent with the contents of the input image. Aerial Diffusion is unable to generate good quality images of scenes that have many objects. Our method is able to deal with complex scenes as well as modify the viewpoint.

We show the quantitative results in Figure 8. Our method achieves a higher CLIP score than all prior work, indicating that it is able to generate an aerial view of the scene with contents dictated by the text better than prior work. The A-CLIP score achieved by our method is higher than that of DreamBooth and Imagic, indicating better conformance to the aerial viewpoint. Even though the A-CLIP score of HawkI is lower than than of Aerial Diffusion, Aerial Diffusion generates poor quality images for scenes with more than one object and also has diagonal artifacts in its generated images, as we observe from the other metrics and qualitative results, thus, offsetting its high A-CLIP Score.

CLIP-I and self-supervised metrics such as SSCD and DINO are not viewpoint invariant. In many cases, since Imagic and DreamBooth generate views close to the input view, rather than aerial views, it is natural for them to have higher CLIP-I, SSCD and DINO Scores. Our method has a much higher CLIP-I, SSCD and DINO score than Aerial Diffusion, showing considerable improvement over prior work in retaining the fidelity and 3D consistency w.r.t. the input image, while modifying the viewpoint. In summary, our specialized aerial-view synthesis method achieves the best viewpoint-fidelity trade-off amongst all related prior work.

## 4.2 Comparisons against 3D based novel view synthesis (NVS) methods

We compare with state-of-the-art benchmark methods on stable diffusion based novel view synthesis from a single image in Figure 5. Zero-1-to-3(Liu et al., 2023b) and Zero123++ (Shi et al., 2023a), both, train on large amounts of multi-view and 3D data from Objaverse (Deitke et al., 2023) contain $800K+$ 3D models; in addition to leveraging a pretrained text-to-2Dimage stable diffusion model. In contrast, our method does not use any multi-view or 3D information and is capable of generating better results in multiple cases. Another task-level difference between our method and prior work on NVS is that the latter aim to explicitly control the camera angle and generate 3D objects in Zero-1-to-3(Liu et al., 2023b), the camera-angle generated by our method is arbitrary within the realms of the text control. The CLIP scores on HawkI-Syn for Zero123++ and HawkI are 0.3071 and 0.3115 respectively, the DINO scores are 0.4341 and 0.4466 respectively. On HawkI-Real, the CLIP score for Zero123++ and HawkI are 0.2908 and 0.3077 respectively, the DINO scores are 0.3916 and 0.3956 respectively. Our aerial-view synthesis method, even without any 3D/ multi-view information and large dataset training, is better than or comparable to 3D-based NVS methods.

## 4.3 Downstream application.

We performed a proof-of-concept experiment using HawkI to generate synthetic images for key-frames of scene-based human actions in the UAV Human (Li et al., 2021) dataset. By computing the L2 distance between self-supervised features of UAV Human video frames and synthetic images on actions: 'smoking',

'pushing someone', 'high five' and 'walking', we achieved a 32.14% accuracy in zero-shot action recognition, an (absolute) improvement of 7.14%.

### 4.4 Ground-truth comparison

We obtained 3D models and text descriptions from Dreamfusion (Poole et al., 2022) to extract the front-view and top-view (ground-truth or GT). We evaluated Zero123++ and HawkI using CLIP Scores (higher the better), and the numbers were 0.2991, 0.3316, respectively. The DINO score (higher the better), which measures self-supervised similarity between the generated images and the GT, are 0.3912, 0.3710 for Zero123++ and HawkI respectively. The LPIPS scores (lower the better) are 0.5801, 0.6373 for Zero123++ and HawkI respectively - our method generates images that are higher in elevation due to 'aerial view' being the text-control. Overall, our 3D-free method, built on stable diffusion, is comparable to 3D methods such as Zero123++ which uses stable diffusion + 800k+ 3D objects.

### 4.5 Ablation studies

We show ablation experiments in Figure 5. In the second column, we show results of our model where it is neither finetuned on the homography image nor uses mutual information guidance for sampling. Thus, the text embedding for the input image, followed by the diffusion UNet are finetuned and the diffusion model generates the aerial image by diffusion denoising, without any mutual information guidance. Many of the generated images either have low fidelity or have low correspondence to the aerial viewpoint. In the experiment in the third column, we add mutual information guidance to the model in column 2. We see higher fidelity (than column 2) of the generated images w.r.t. the input image. In the fourth column, we add the homography image finetuning step to the model in column 2, but do not use mutual information guidance at inference. The generated images, in many cases, are aerial, but have lower fidelity w.r.t. the input image. In the final column, we show results with our full model. The generated images achieve the best trade-off between the viewpoint being aerial and fidelity w.r.t. the input image, in comparison to all ablation experiments. Quantitative analysis:

- **Effect of $I_H$:** To study the effect of $I_H$, we compare the CLIP and A-CLIP scores for the full model vs full model w/o $I_H$. The scores in all cases where $I_H$ is not present are lower, indicating lower consistency with the viewpoint being 'aerial'. For instance, on HawkI-Real, the CLIP scores for the full model and full model w/o $G_{MI}$ are 0.3077 and 0.3040 respectively. The A-CLIP scores for the full model and full model w/o $I_H$ are 0.1887 and 0.1842 respectively.
- **Effect of $G_{MI}$:** To study the effect of $I_H$, we compare the SSCD and DINO scores for the full model vs model w/o $G_{MI}$. The scores in all cases where $G_{MI}$ is not present are lower. For instance, on HawkI-Real, the SSCD scores for the full model and model w/o $G_{MI}$ are 0.3314 and 0.3204 respectively. The DINO scores for the full model and model w/o $G_{MI}$ are 0.3956 and 0.39 respectively.

### 4.6 Comparison with other metrics for guidance.

We compare with two other metrics for diffusion guidance at inference: (i) L2 distance between the features of the generated image and the input image, (ii) a metric inspired by Wasserstein distance or Earth Mover's distance, for which we compute the distance between the histograms of the probability distributions of the two images. Our mutual information guidance method is *better at preserving the fidelity w.r.t the input image*, as evidenced by *higher SSCD scores*. The SSCD score on HawkI-Real for Wasserstein guidance, $L2$ guidance, and mutual information guidance are 0.3137, 0.3204 and 0.3314 respectively. The DINO score on HawkI-Real for Wasserstein guidance, $L2$ guidance, and mutual information guidance are 0.3847, 0.3858 and 0.3956 respectively.

### 4.7 Text controlled view synthesis: Other views

HawkI can be extended to generate other text-controlled views such as side view, bottom view and back view (Figure 20). We modify the target text $t_T$ to indicate different viewpoints, and retain the other hyperparameters.

### 4.8 3D-free HawkI + 3D priors?

Our 3D-free approach **complements** 3D-based methods (Shi et al., 2023a; Lin et al., 2023). While **3D data collection and large-scale training are expensive and unsustainable**, front-view 2D images

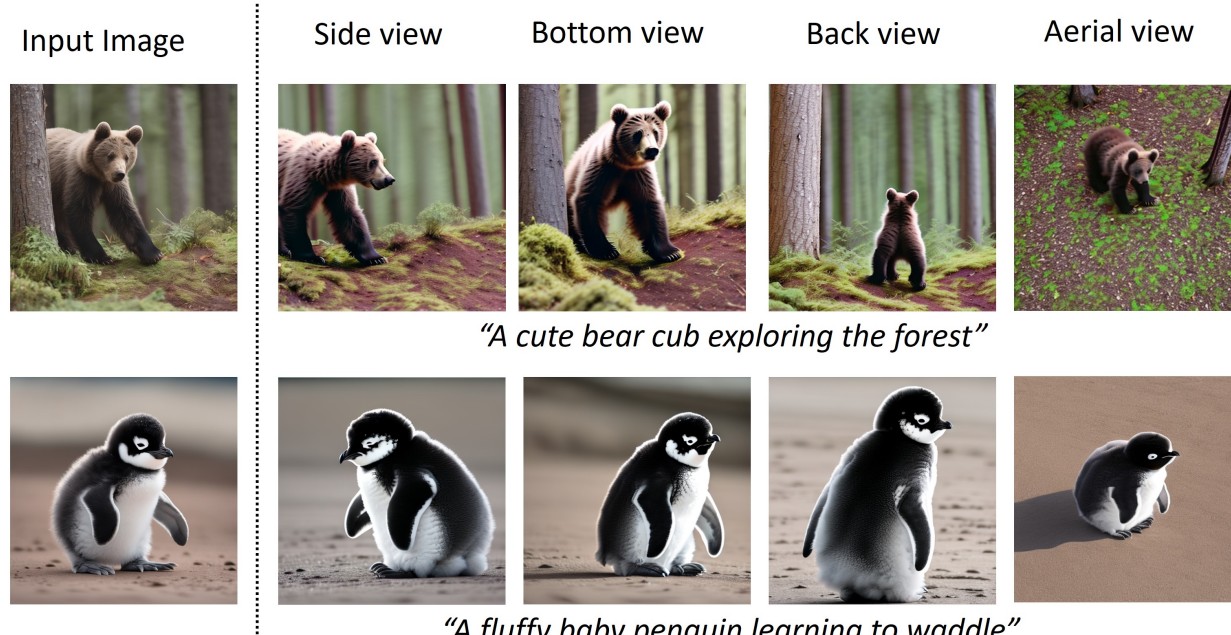

Figure 6: HawkI can be extended to generate other text-controlled views as well.

are more readily available[35]. Hence, it is beneficial to solve the fundamental problems associated with the task in a data-efficient (or 3D-free) manner. Our goal is to push the frontiers of 3D-free aerial-view generation from a single image, achieving results comparable to methods (Shi et al., 2023a) using stable diffusion + 800k 3D objects. That being said, HawkI be combined with 3D approaches, to multiply the benefits of 3D and 3D-free approaches. To validate this, we replaced the homography prior with the image from the 3D-based Zero123++ (Shi et al., 2023a) approach and evaluated our HawkI model on images from HawkI-Syn. The CLIP scores for Zero123++, HawkI, and "HawkI with Zero123++ prior" are 33.17, 33.17, and 33.81, respectively. The DINO scores for Zero123++, HawkI, and "HawkI with Zero123++ prior" are 0.3613, 0.3977, and 0.4612, respectively, thus demonstrating our claim!. Also, using 3D priors with HawkI allows finer camera control.

**More results.** Please refer to the supplementary material for (i) more qualitative comparisons with text + exemplar image and 3D based NVS methods, (ii) qualitative comparisons with other guidance metrics, (iii) detailed quantitative results for ablations, (iv) comparison of IPM with data augmentation, (v) more results on other text-controlled views, (vi) qualitative comparisons with warping + outpainting (scene extrapolation) and ControlNet variations.

## 5 Conclusions, Limitations and Future Work

We present a novel method for aerial view synthesis. Our goal is to leverage pretrained text-to-image models to advance the frontiers of text + exemplar image based view synthesis *without* any additional 3D or multi-view data at train/ test time. Our method has a few limitations, which form an avenue for future work - (i) Combining our 3D-free approach with 3D priors can enable the generation of camera-controlled views. (ii) Using our plug-and-play method with stronger text-to-image backbone models can reduce hallucination and improve fidelity to input images. Other directions for future work are - (i) IPM is crucial for aerial view generation, allowing exploration of similar camera models and homography projections for various scenes, (ii) our mutual information guidance can be applied to other image editing and personalization tasks, (iii) generated data can support UAV and aerial-view applications like cross-view mapping, 3D reconstruction, and synthetic data tasks.

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

**Why is a 3D-free approach important?** Collecting and training on 3D data at a large scale is expensive and unsustainable. Even with 3D datasets like Objaverse or MVImageNet, the data is specific to certain scenes and objects. While training on such data can yield good performance for those specific scenarios, it offers limited generalization to other in-the-wild or out-of-distribution scenes. This limitation is due to the relatively small size of these datasets (around 800k objects), and scaling to the level of image datasets like LAION-5B for 3D objects is prohibitively expensive and challenging.

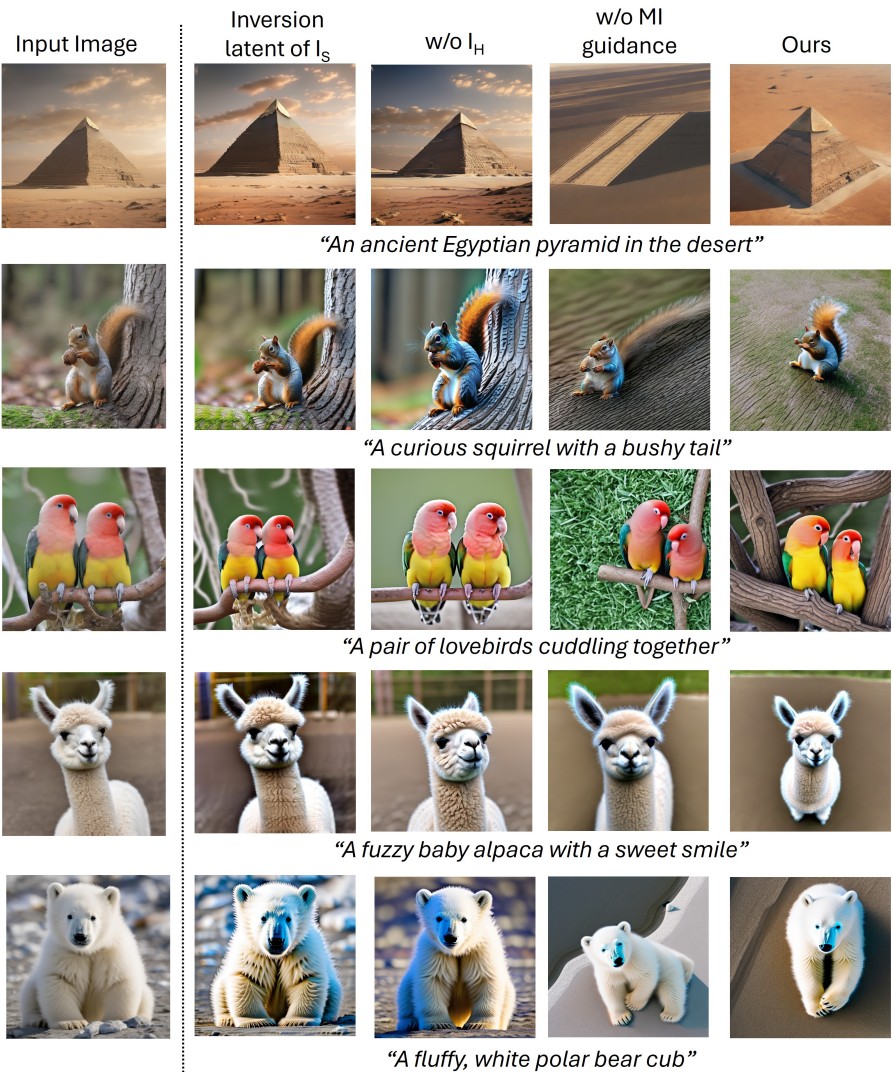

Figure 7: We show more qualitative results for ablation experiments. Our ablation on using the inversion latent of $I_S$ with a target prompt to perform this task are presented in Column 2. Without the multi-step optimization process, which is necessary for achieving an appropriate bias-variance trade-off while learning the characteristics of the input image, there are high bias issues, and the model fails to generate the aerial-view image. Without the homography image optimization step (column 3), the model fails to produce an aerial-view image. Guidance with the homography image is crucial for viewpoint translation. The mutual information guidance formulation enhances the fidelity of the generated aerial-view images relative to the input image. In summary, our optimization step, which involves fine-tuning with the homography image, along with the mutual information guidance formulation, holistically generates a high-fidelity aerial-view image.

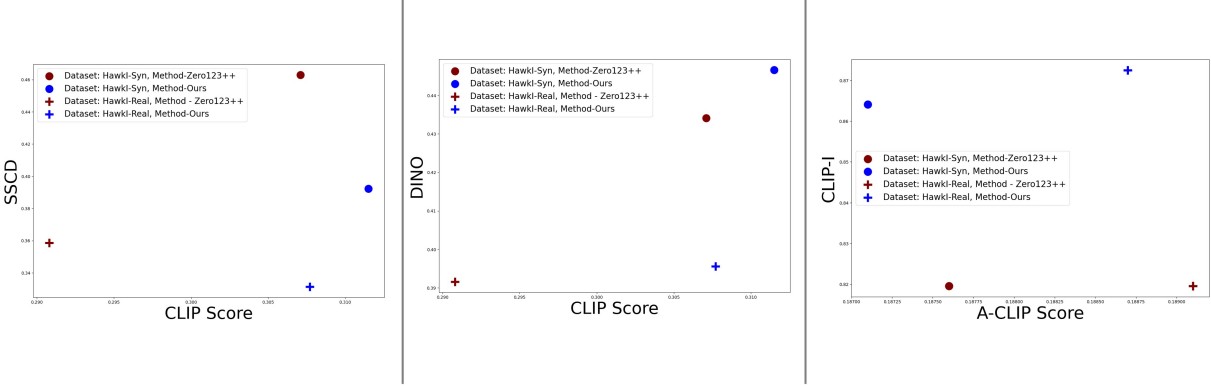

Figure 8: We show detailed quantitative comparisons of HawkI against Zero123++, a state-of-the-art 3D-based novel-view synthesis method. Zero123++ uses 800k+ 3D objects in its finetuning of a stable diffusion model. In contrast, our method, HawkI, uses absolutely **no 3D information** at test-time finetuning of the stable diffusion model or at inference; and is able to achieve **comparable or better performance** on various metrics indicative of viewpoint or fidelity w.r.t. input image. Moreover, since HawkI performs 3D-free test-time optimization + inference on a pretrained stable diffusion model, it is easily applicable to any in-the-wild image without any additional generalization issues or constraints, beyond the pretrained stable diffusion model itself.

| Method | CLIP | A-CLIP | SSCD | DINO | CLIP-I |
|---|---|---|---|---|---|
| Dataset: HawkI - Syn | | | | | |
| w/o MI, w/o $I_H$ | 0.3112 | 0.1832 | 0.4609 | 0.5042 | 0.8839 |
| w/o MI | 0.3109 | 0.1861 | 0.3900 | 0.4481 | 0.8673 |
| w/o $I_H$ | 0.3112 | 0.1834 | 0.4570 | 0.5016 | 0.8820 |
| Ours ($\lambda_{MI} = 1e - 5$) | 0.3115 | 0.1857 | 0.3922 | 0.4466 | 0.8664 |
| Ours ($\lambda_{MI} = 1e - 6$) | 0.3114 | 0.1871 | 0.3860 | 0.4427 | 0.8641 |
| Dataset: HawkI - Real | | | | | |
| w/o MI, w/o $I_H$ | 0.3048 | 0.1848 | 0.4013 | 0.4391 | 0.8922 |
| w/o MI | 0.3047 | 0.1885 | 0.3204 | 0.3900 | 0.8721 |
| w/o $I_H$ | 0.3040 | 0.1842 | 0.4000 | 0.4470 | 0.8921 |
| Ours ($\lambda_{MI} = 1e - 5$) | 0.3038 | 0.1861 | 0.3284 | 0.3941 | 0.8747 |
| Ours ($\lambda_{MI} = 1e - 6$) | 0.3077 | 0.1887 | 0.3314 | 0.3956 | 0.8725 |

Table 1: We report the quantitative metrics for the ablation experiments corresponding to removing the $I_H$ finetuning step, removing mutual information guidance or removing both. We use two additional quantitative metricsn - CLIP-D and A-CLIP-D which analyze directional similarity. CLIP directional similarity docs measures the consistency of the change between two images, $I_S$ and $I_T$, in the CLIP space with the change between the two image captions (dictating the transformation from $I_S$ to $I_T$ ). We compute two versions of this score, CLIPD score and the A-CLIPD score. CLIPD score uses 'aerial view, ' + text as the target text, and 'A-CLIPD score' uses 'aerial view' as the target text. Without finetuning on $I_H$, while the generated images have high fidelity w.r.t the input image, the generated images score low on the aspect of the viewpoint being aerial, adding the $I_H$ finetuning step enables the generation of 'aerial' images. Without mutual information guidance, the generated images have low fidelity w.r.t. the input images, adding mutual information guidance steers the content in the generated image towards the content in the input image. In summary, our full model, with the inverse perspective mapping finetuning step as well as mutual information guidance, achieves the best viewpoint-fidelity trade-off amongst all ablation experiments.

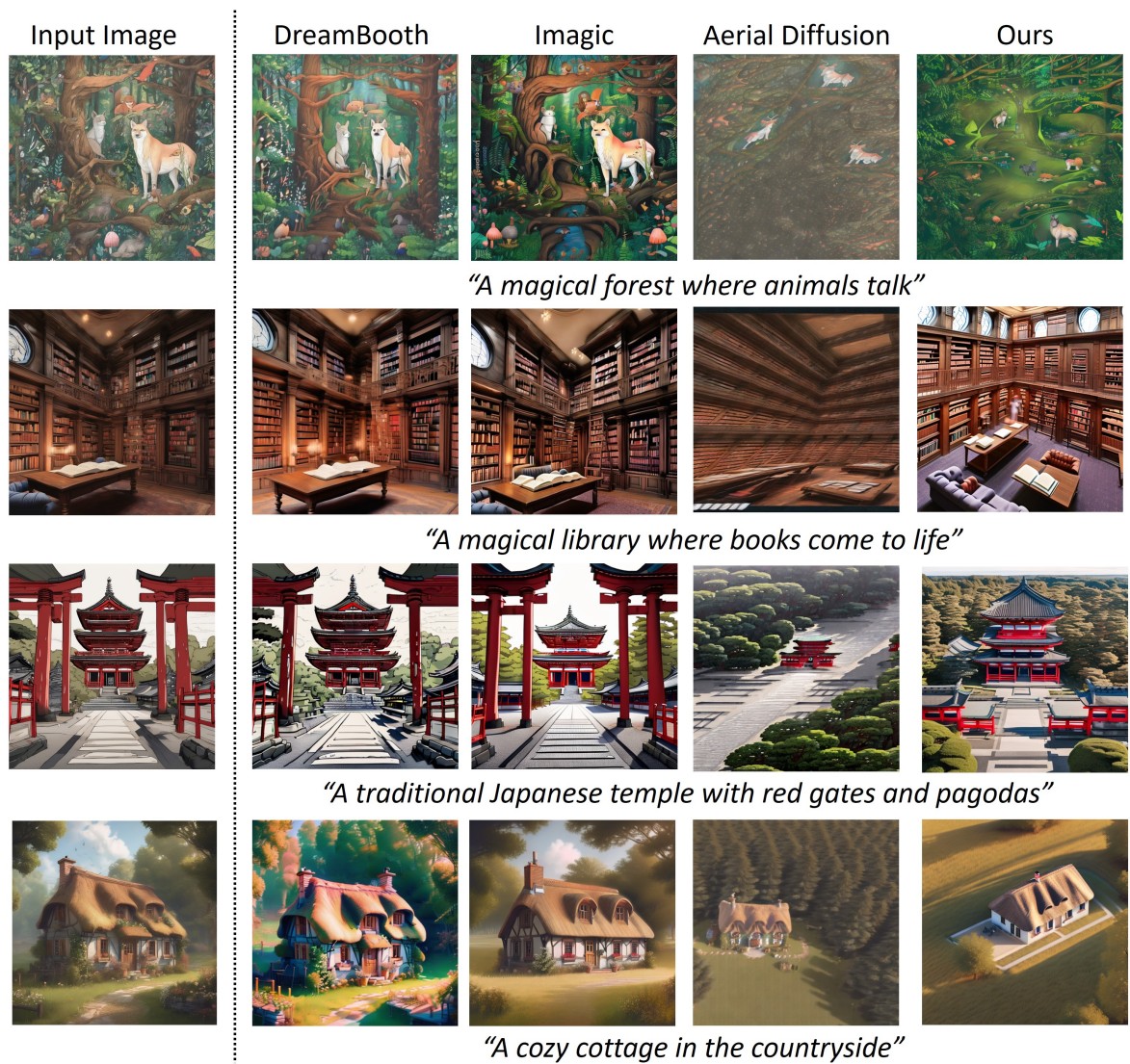

Figure 9: Compared to state-of-the-art text + exemplar image based methods, HawkI is able to generate images that are "more aerial", while being consistent with the input image. The images are from the HawkI-Syn dataset.

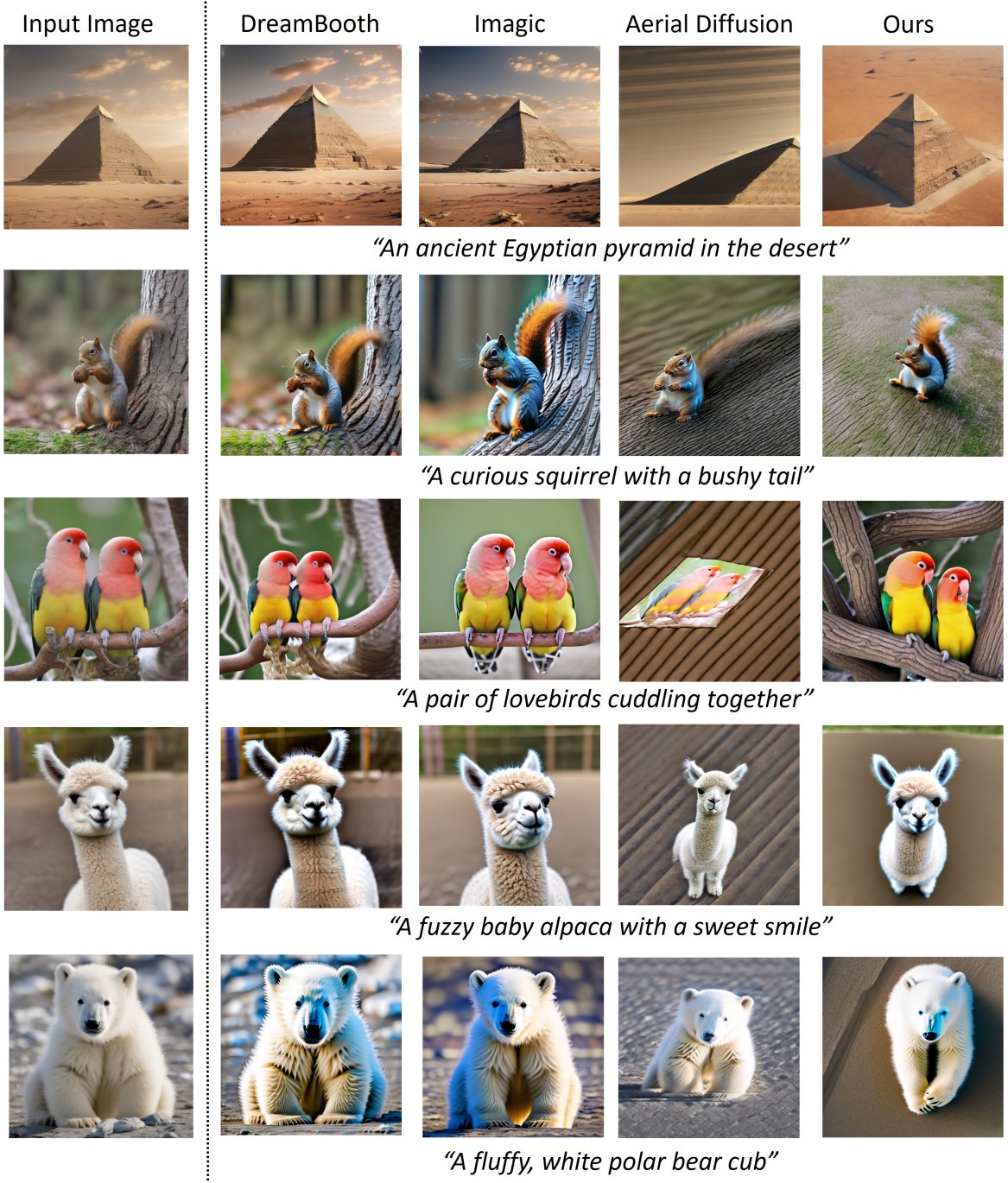

Figure 10: Compared to state-of-the-art text + exemplar image based methods, HawkI is able to generate images that are "more aerial", while being consistent with the input image. The images are from the HawkI-Syn dataset.

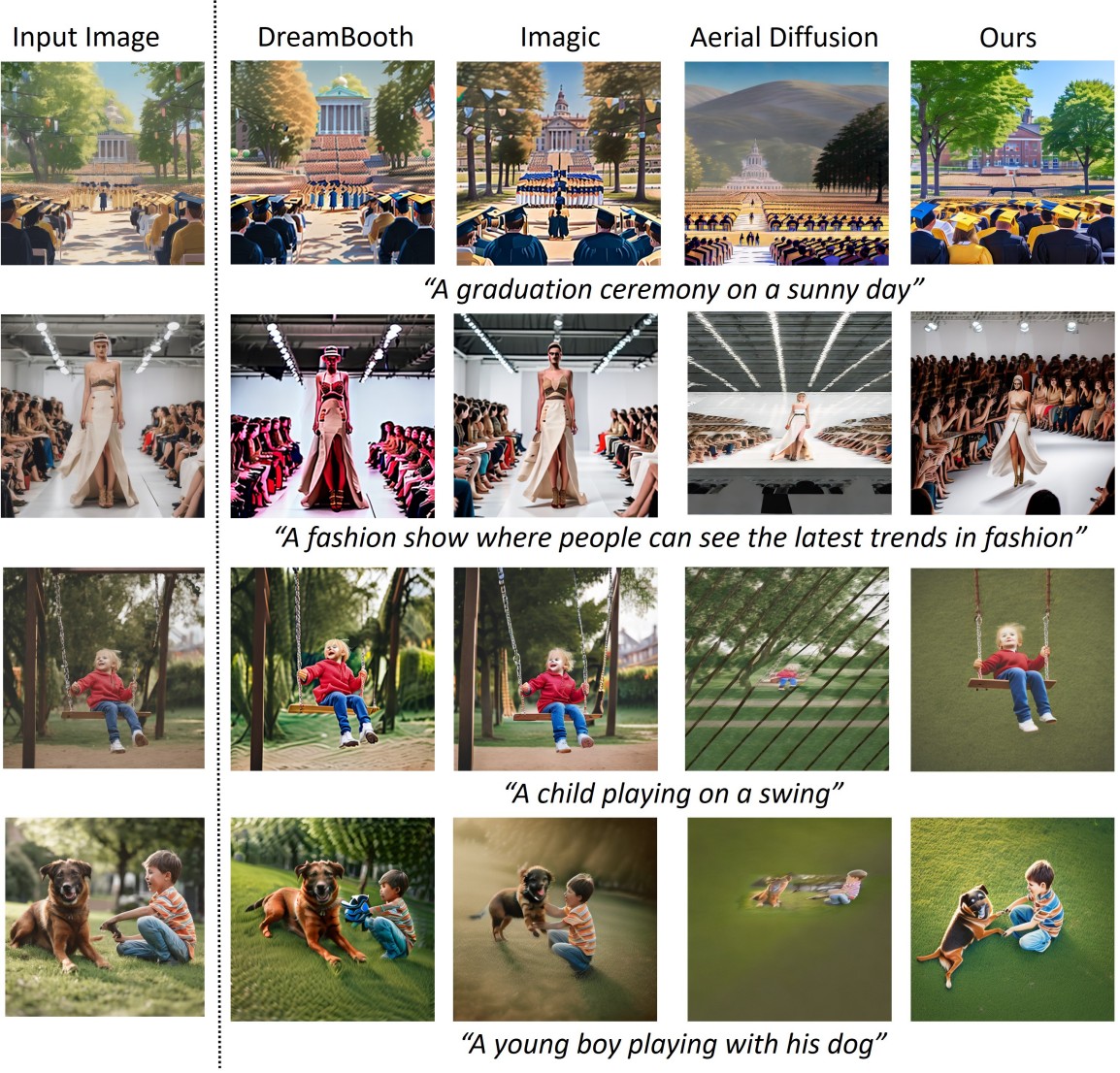

Figure 11: Compared to state-of-the-art text + exemplar image based methods, HawkI is able to generate images that are "more aerial", while being consistent with the input image. The images are from the HawkI-Syn dataset.

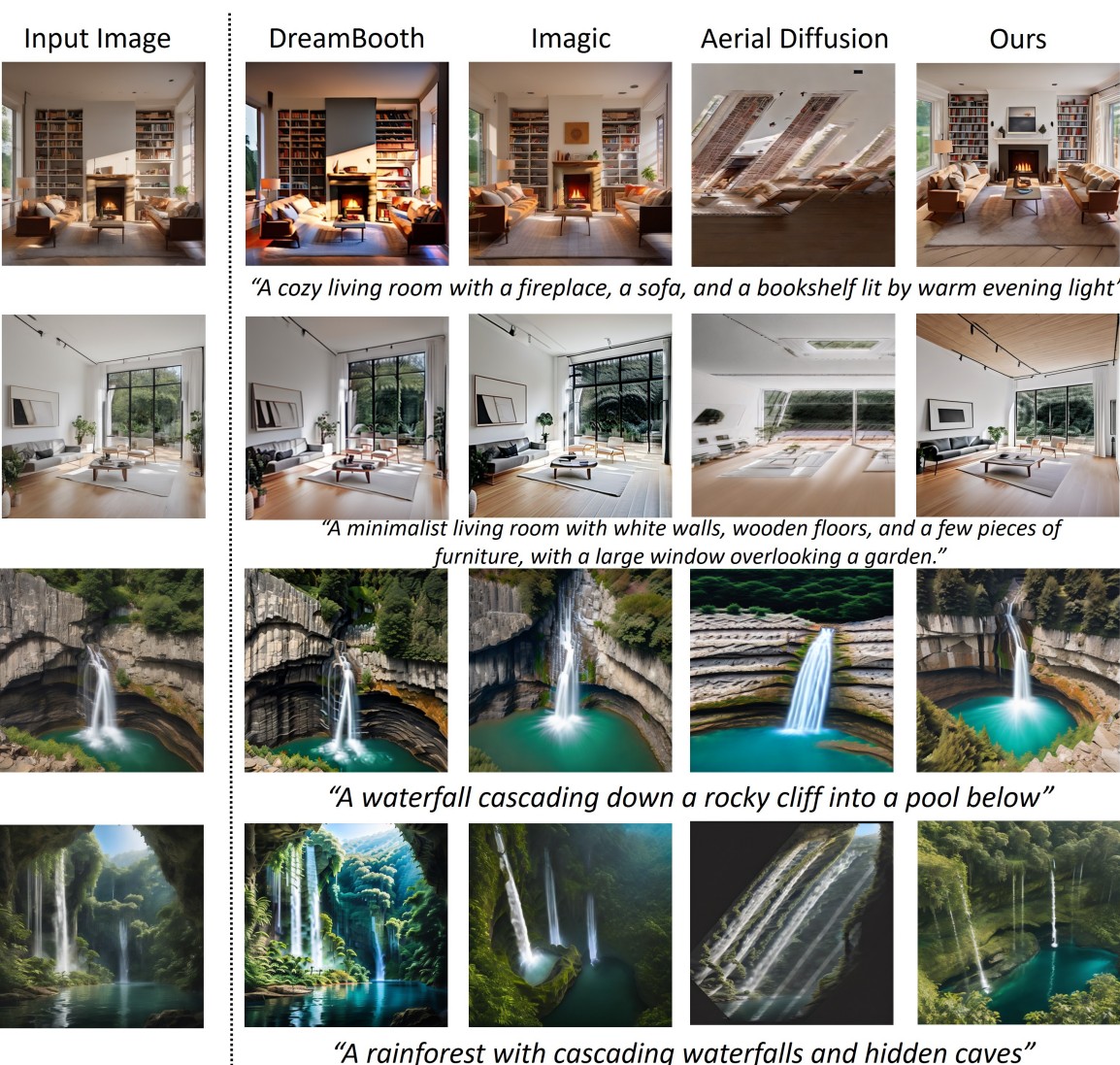

Figure 12: Compared to state-of-the-art text + exemplar image based methods, HawkI is able to generate images that are "more aerial", while being consistent with the input image. The images are from the HawkI-Syn dataset.

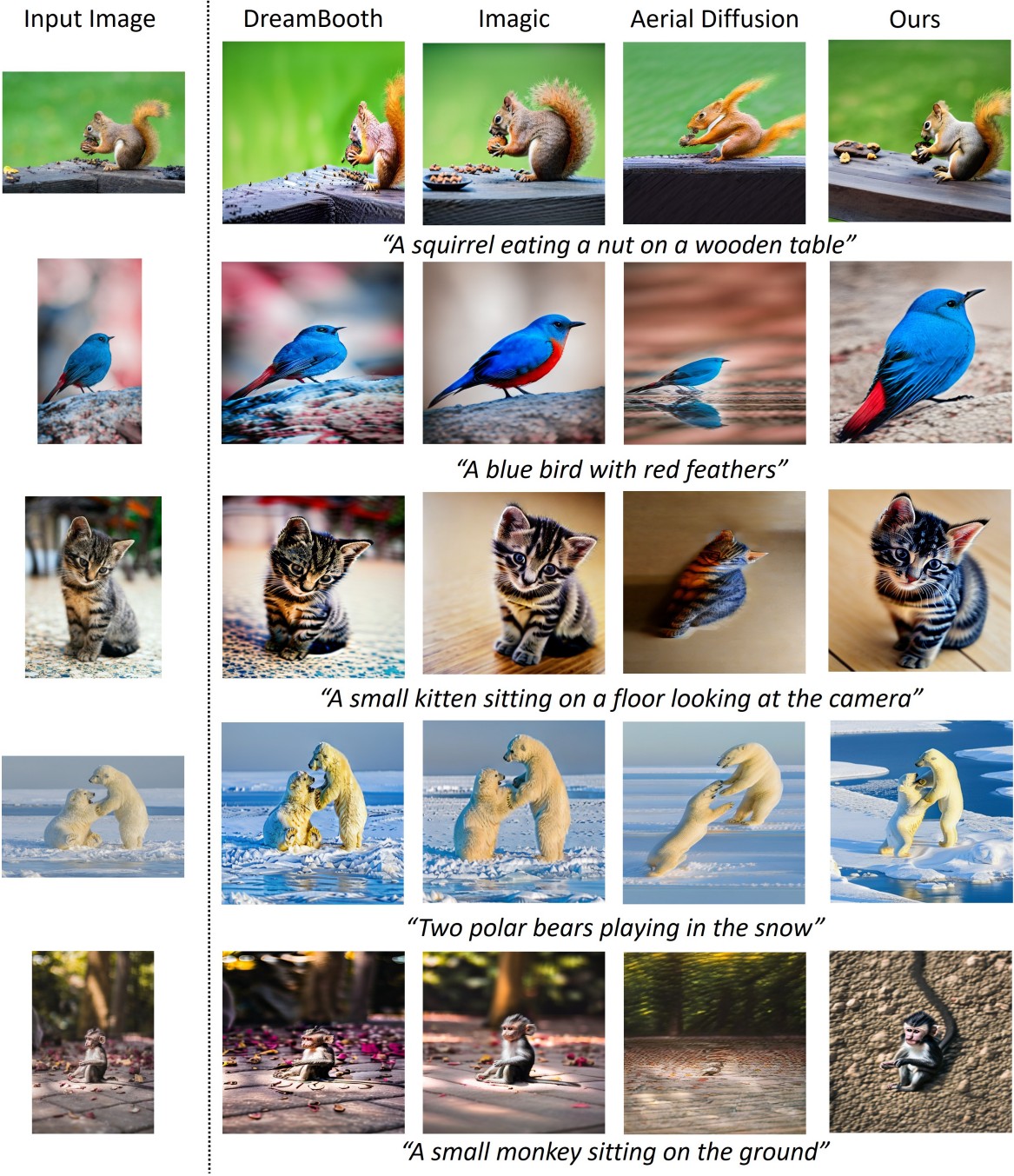

Figure 13: Compared to state-of-the-art text + exemplar image based methods, HawkI is able to generate images that are "more aerial", while being consistent with the input image. The images are from the HawkI-Real dataset.

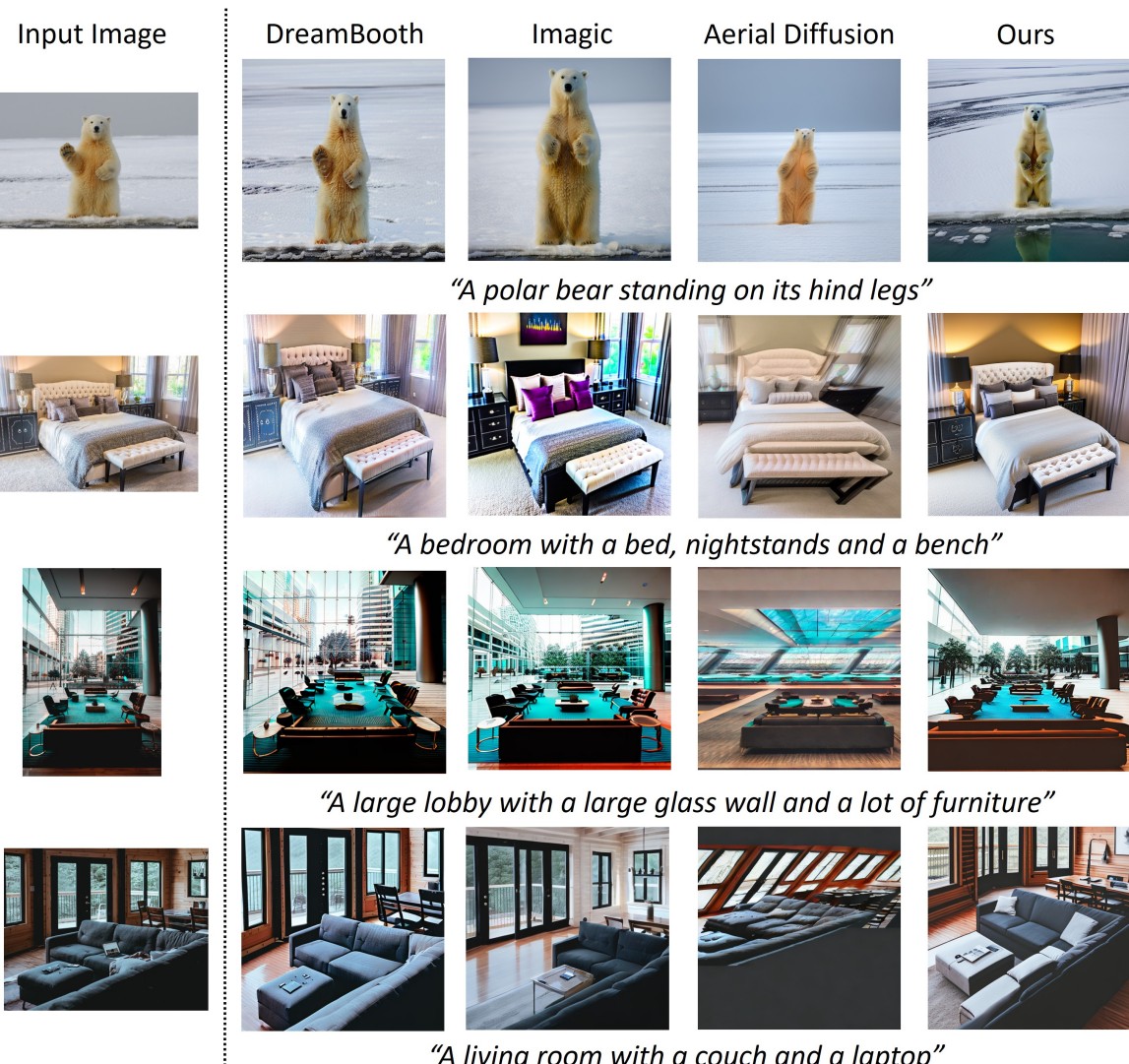

Figure 14: Compared to state-of-the-art text + exemplar image based methods, HawkI is able to generate images that are "more aerial", while being consistent with the input image. The images are from the HawkI-Real dataset.

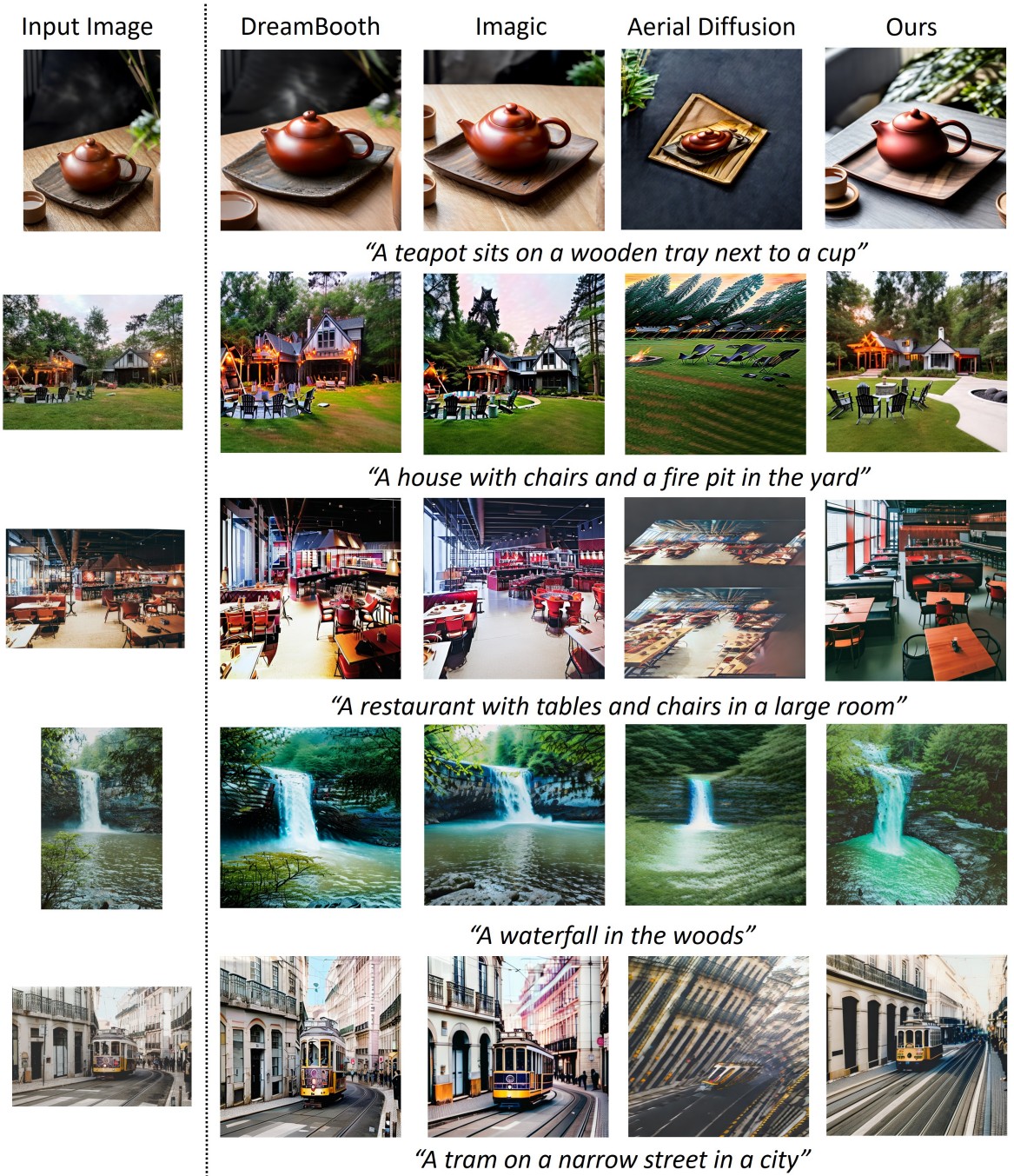

Figure 15: Compared to state-of-the-art text + exemplar image based methods, HawkI is able to generate images that are "more aerial", while being consistent with the input image. The images are from the HawkI-Real dataset.

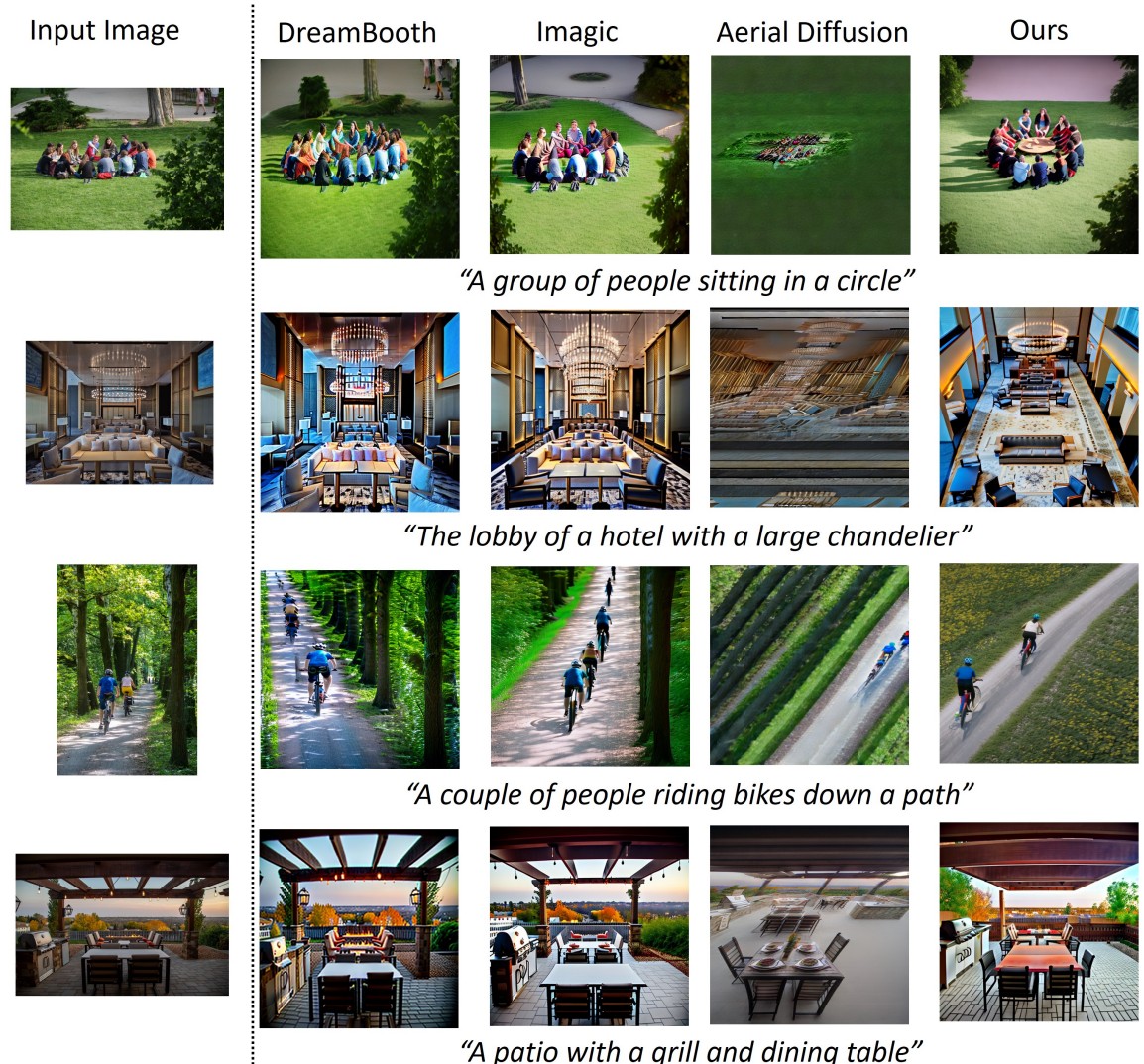

Figure 16: Compared to state-of-the-art text + exemplar image based methods, HawkI is able to generate images that are "more aerial", while being consistent with the input image. The images are from the HawkI-Real dataset.

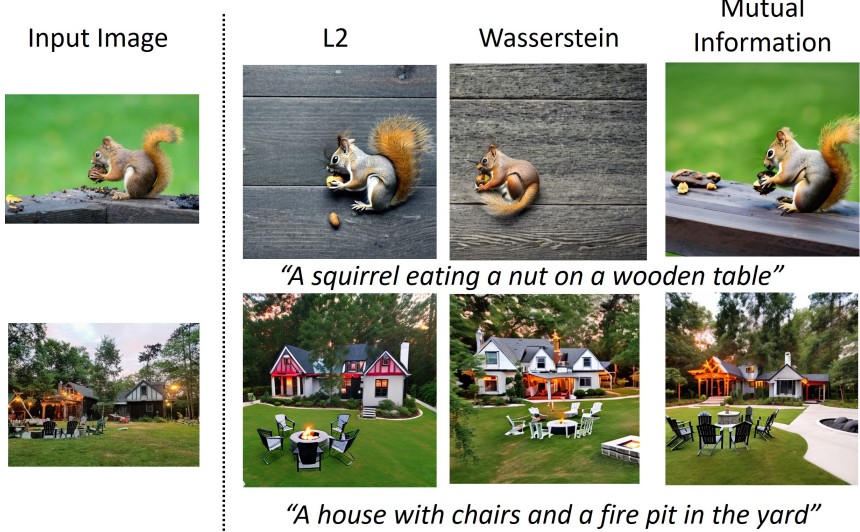

Figure 17: We show a few examples for comparisons with other metrics for diffusion guidance such as L2 distance and Wasserstein distance. Our mutual information guidance method is better at preserving the fidelity w.r.t the input image, as also evidenced by higher SSCD scores. The SSCD score for Wasserstein guidance, $L2$ guidance, and mutual information guidance are 0.3181, 0.3224 and 0.3345 respectively, averaged over all images in the HawkI-Real dataset.

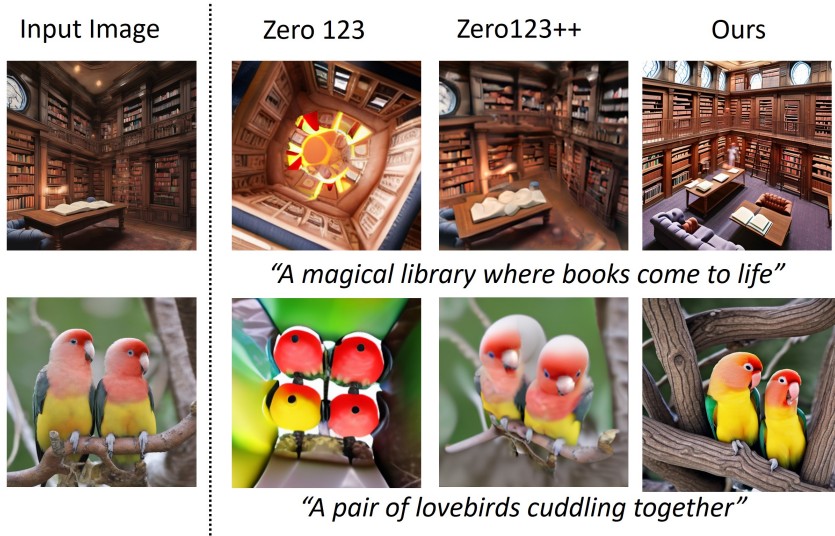

Figure 18: We compare with latest related work on novel view synthesis: Zero-1-to-3 and Zero123++ on images from HawkI-Syn. Both of these methods use the pretrained stable diffusion model and the 3D objects dataset, Objaverse with 800k+ 3D objects, for training. Our method uses just the pretrained stable diffusion model for the task of aerial view synthesis from a single image.

3D generation methods like Zero123++ are capable of generating different views with high fidelity by using pretrained stable diffusion models to finetune on large-scale 3D objects datasets. However, their generalization capabilities are limited. Our method is able to generate high quality aerial images for the given input images without any 3D data and using just the pretrained text-to-2D image stable diffusion model, however, there is scope for improving the fidelity of the generated aerial image w.r.t the input image. Moreover, our method controls the viewpoint via text and does not provide the provision to quantitatively control the camera angle. Both of these limitations of our method can be alleviated by exploring the combination of pretrained Zero123++ models (or other 3D models) and our method, as a part of future work.

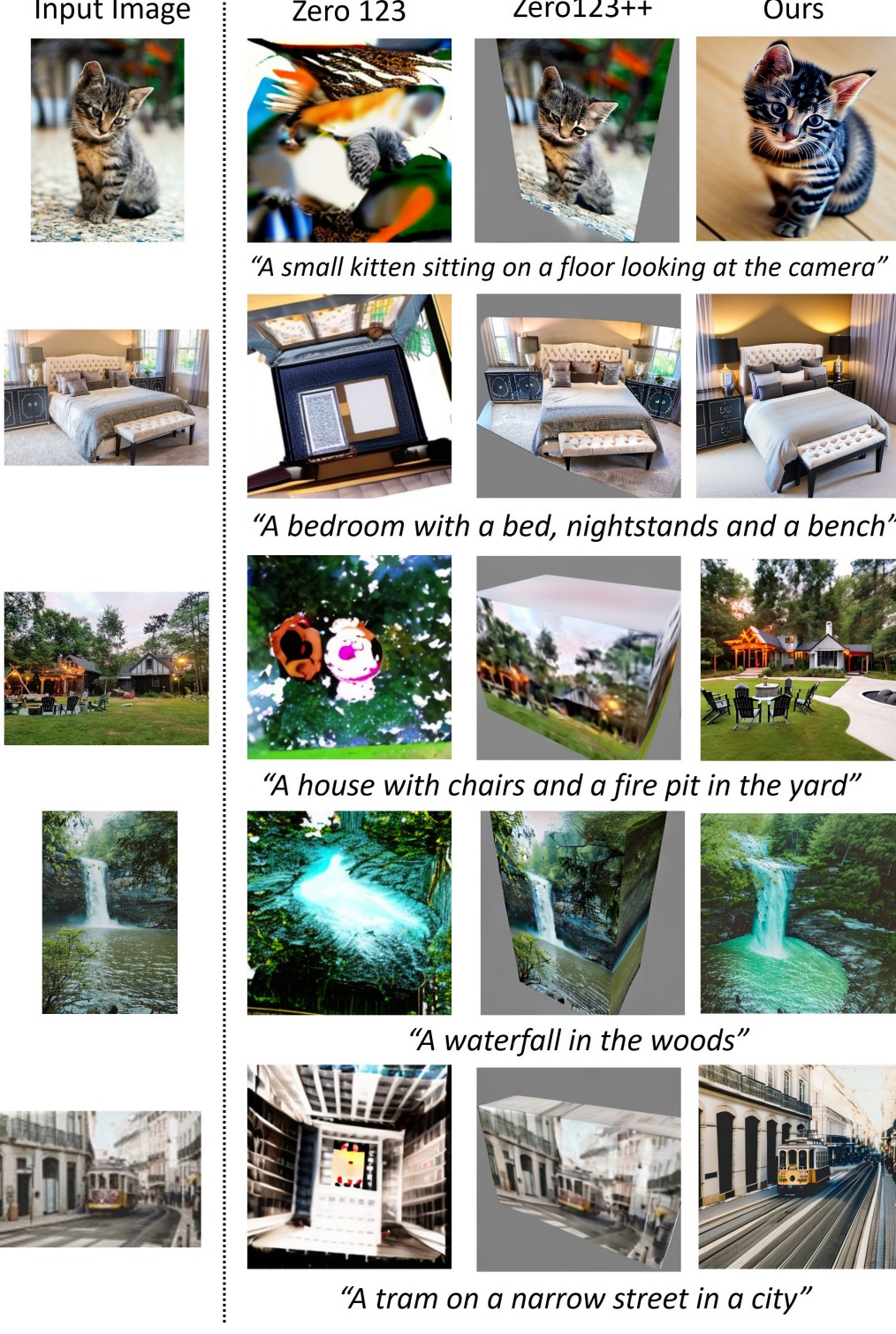

Figure 19: We compare with latest related work on novel view synthesis: Zero-1-to-3 and Zero123++ on images from HawkI-Real. Both of these methods use the pretrained stable diffusion model and the 3D objects dataset, Objaverse with 800k+ 3D objects, for training. Our method uses just the pretrained stable diffusion model for the task of aerial view synthesis from a single image.

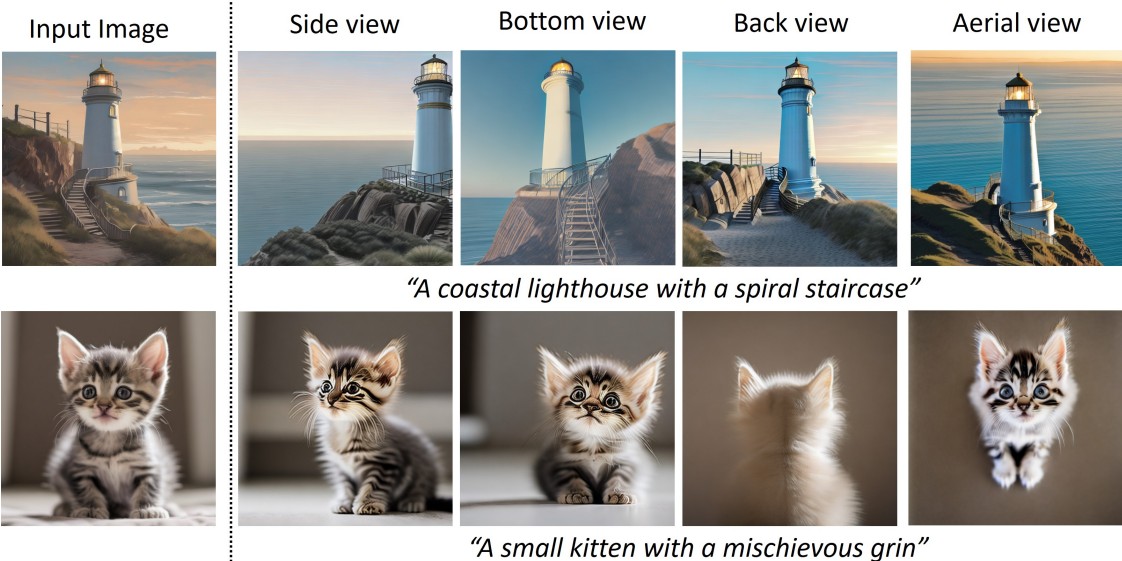

Figure 20: Additional results on extending HawkI to generate other text-controlled views.

On the other hand, front-view 2D images are more readily available, and there are 2D text-image diffusion models trained on billions of data points. Hence, it is advantageous to tackle fundamental problems in a data-efficient (or 3D-free) manner using only 2D information. In this paper, our goal was to push the boundaries of 3D-free aerial-view generation from a single image, achieving results comparable to methods that use stable diffusion with 800k 3D objects.

As a subsequent objective, we explored using pre-trained 3D models. Although 3D models provide enhanced 3D controllability, their generalizability is limited due to data scale issues. Thus, we ask the following question: Can HawkI be combined with 3D approaches to take advantage of the benefits of both 3D and 3D-free methods? We found that this combination harnesses the data efficiency and generalizability of HawkI while generating aerial views of any in-the-wild image with the 3D control benefits of pretrained 3D models from existing Objaverse datasets, which may not be broadly generalizable. This paves the way for new research directions.

**Sensitivity to hyperparameters.** Our proposed method is not sensitive to hyperparameters per image. In fact, we do not finetune any hyperparameters for individual images. We used a small validation subset of images to set the hyperparameters and have since used the same values for all images without any changes. To select hyperparameters, we used the same parameters as Aerial Diffusion for optimizing the text embedding and fine-tuning the diffusion UNet on the input image. For the homography image, we halved the number of iterations while retaining all other parameters. The only hyperparameter requiring consideration was mutual information, which we tuned using the validation subset. Thus, there is no per-task or per-image hyperparameter tuning involved.

**Metrics (Definition).**

CLIP Score: The CLIP (Contrastive Language-Image Pre-training) score is a metric used to evaluate the similarity between an image and a text description. It measures how well a generated caption matches the actual content of an image1. The score is calculated using the cosine similarity between the embeddings of the image and the text, with values ranging from 0 to 1, where higher scores indicate better alignment. The A-CLIP or 'aerial CLIP' score measures the alignment of the generated image with the text caption corresponding to "aerial view", evaluating viewpoint correctness.

CLIP-I Score: This metric is similar to the CLIP score but focuses specifically on the generated images.

DINO Score: The DINO score is a metric used in self-supervised learning for evaluating the performance of models trained without labeled data. It measures the consistency of predictions made by the model compared to the input image, encouraging the high similarity or high fidelity.

SSCD Score: The SSCD (Self-Supervised Copy Descriptor) metric is used in the context of image copy detection. It evaluates the effectiveness of a model in identifying copies of images by comparing descriptor

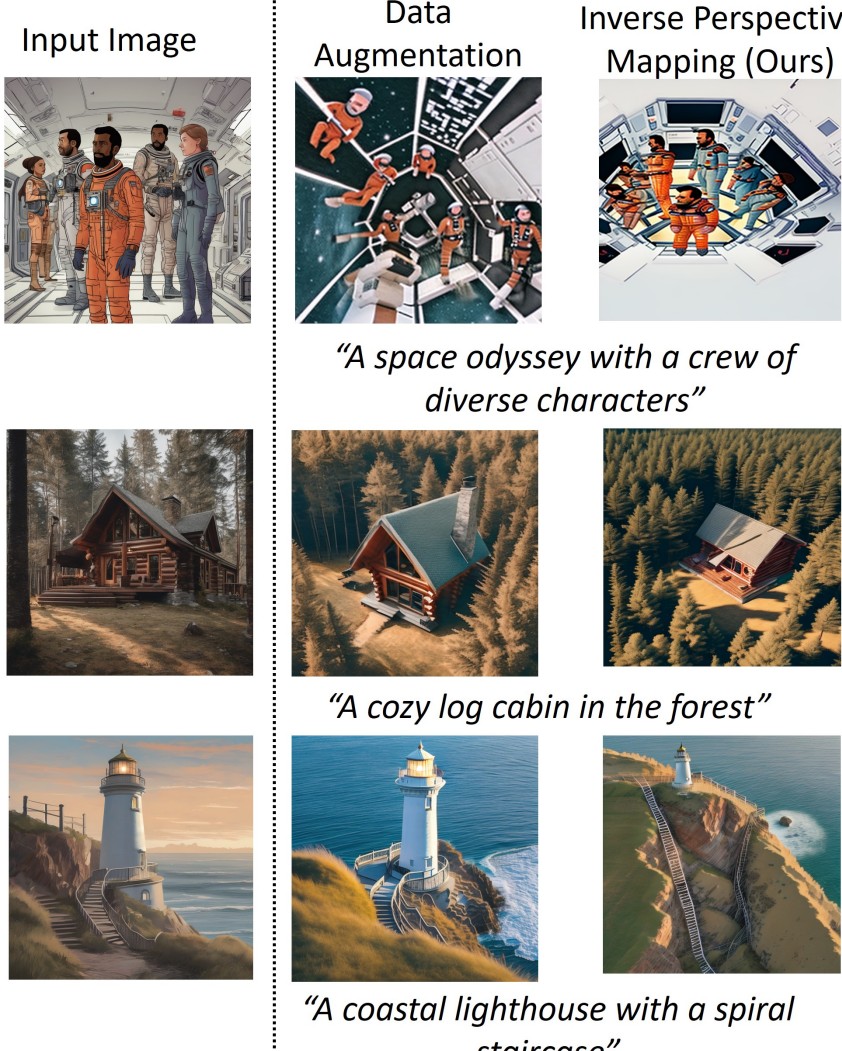

Figure 21: **Can any data augmentation be used in place of Inverse Perspective Mapping (IPM)?** One question that arises from the usage of Inverse Perspective Mapping is related to whether it actually provides pseudo weak guidance, in addition to increasing variance (or reducing bias) in the representation space that is being conditioned for aerial view generation. The latter can be achieved with any random data augmentation. To understand this, we use a 45 degrees rotated image in place of the image corresponding to the Inverse Perspective Mapping in the second stage of finetuning the text embedding and the diffusion UNet. We do not use mutual information guidance in any of our experiments, to ensure that our findings are disentangled to the effects of the homography transformation. Our finding is that results with models that use Inverse Perspective Mapping are generally better in terms of the viewpoint being aerial, while preserving the fidelity with respect to the input image, than models that use the 45 degree rotated image. The CLIP scores for the 45-degree rotated image and IPM (/homography) results are 31.90 and 32.70, respectively. Thus, models employing Inverse Perspective Mapping (IPM) tend to yield **better aerial viewpoints** compared to those using 45-degree rotated images, while maintaining fidelity w.r.t. input image. Hence, we conclude that rather than using any random data augmentation technique, it is beneficial to use IPM as it is capable of providing pseudo weak guidance to the model for aerial view synthesis. This finding also paves direction for future work on using carefully crafted homography priors for view synthesis corresponding to different camera angles and viewpoints.

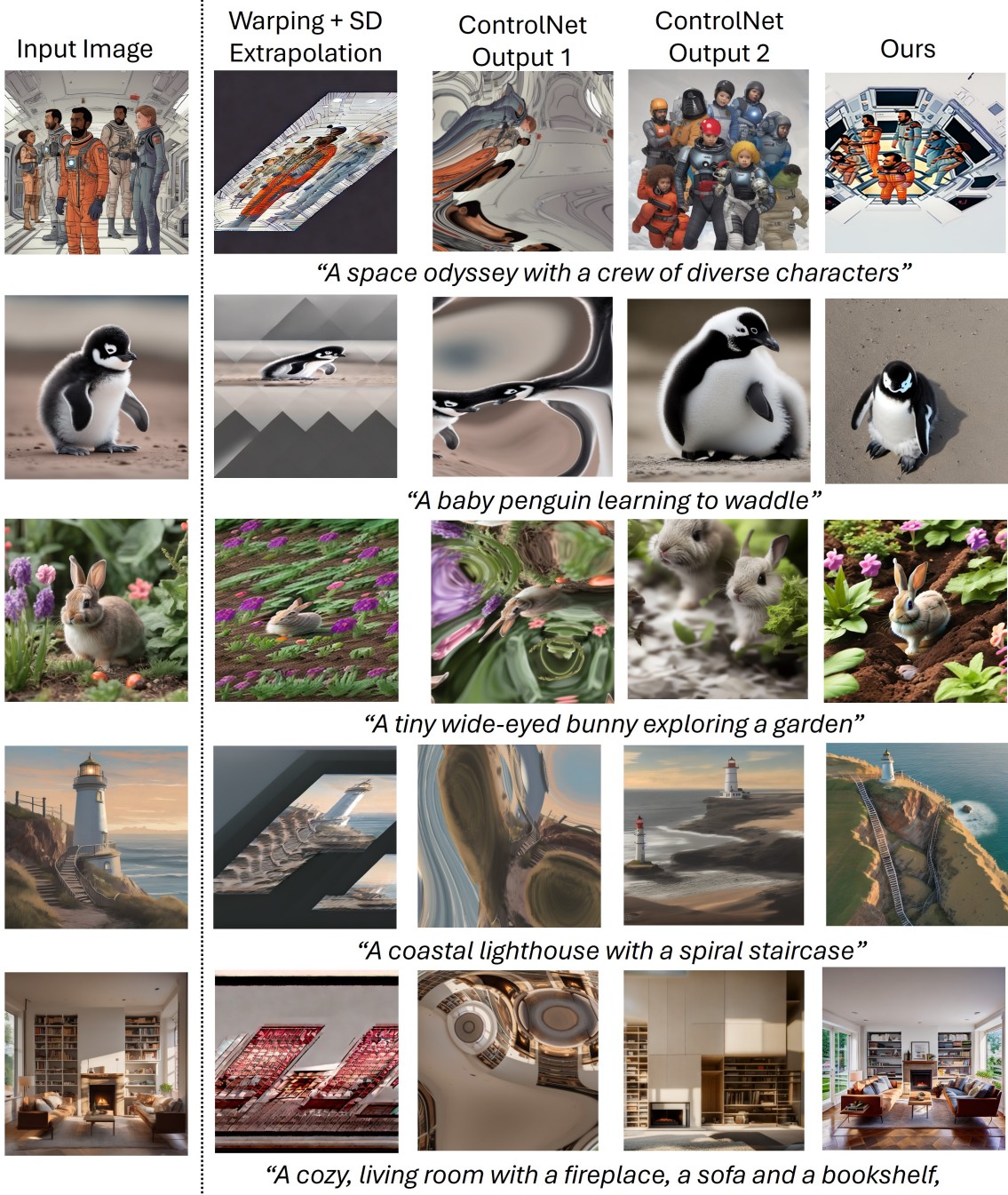

Figure 22: **Comparisons with (i) warping + scene extrapolation, (ii) ControlNet (Zhang et al., 2023).** In the second column, we present results on warping + scene extrapolation. Specifically, we warp the image to its pseudo aerial-view using the IPM, and use Stable Diffusion to extrapolate. To do so, we finetune the Diffusion UNet using the warped image and the text prompt corresponding to 'aerial view' + image description, and run inference using the finetuned diffusion model. Warping + scene extrapolation is highly ineffective, due to the poor quality of pseudo aerial-view images. Our method, HawkI is able to generatefar higher quality images. In the third and fourth columns, we show results with ControlNet Img2Img (https://stablediffusionweb.com/ControlNet). We provide the input image and the text prompt corresponding to 'aerial view' + image description and we show results corresponding to two runs of the model. Typically, ControlNet is highly successful in text-based image to image synthesis in cases dictating small-scale pixel-level. However, it is unable to perform view synthesis i.e. it is unable to generate high-fidelity aerial-view images for a given input image. We do not use mutual information guidance in any of our experiments, to ensure that our findings are disentangled to the effects of the homography transformation.

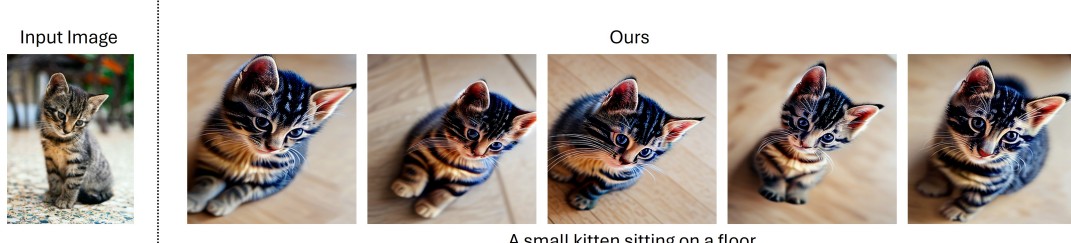

A small kitten sitting on a floor

Figure 23: Our method can generate diverse images for the target viewpoint corresponding to the given input scene.

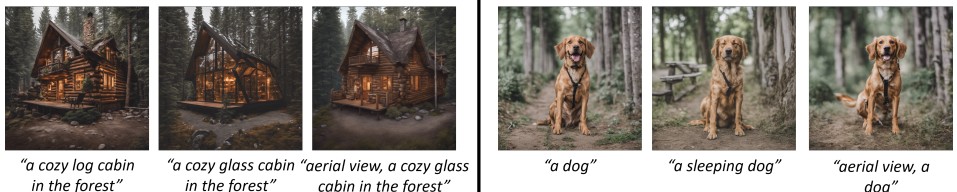

"a cozy log cabin in the forest"    "a cozy glass cabin in the forest"    "aerial view, a cozy glass cabin in the forest"    "a dog"    "a sleeping dog"    "aerial view, a dog"

Figure 24: Methods such as DreamBooth are unable to generate different views corresponding to a specific image though they are able to perform other non-rigid transformations, guided by text. In this image, we show how the model is able to generate a glass cabin, given an image of a log cabin and is able to make a dog that is awake close its eyes. In both cases, the model is unable to generate the aerial view.

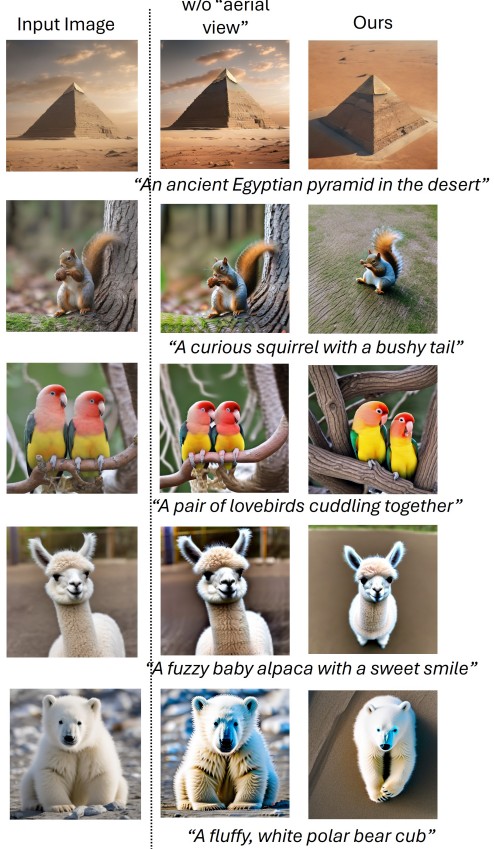

"An ancient Egyptian pyramid in the desert"

"A curious squirrel with a bushy tail"

"A pair of lovebirds cuddling together"

"A fuzzy baby alpaca with a sweet smile"

"A fluffy, white polar bear cub"

Figure 25: **Ablation on using "aerial view" in the target caption.** Without the text guidance corresponding to "aerial view" in the target caption, the model is unable to generate the desired viewpoint.

| Method | CLIP | A-CLIP | SSCD | DINO | CLIP-I |
|---|---|---|---|---|---|
| Dataset: HawkI - Syn | | | | | |
| DreamBooth LoRA | 0.3069 | 0.1765 | 0.6260 | 0.6204 | 0.9058 |
| Imagic | 0.3046 | 0.1748 | 0.5559 | 0.5846 | 0.9064 |
| Aerial Diffusion | 0.2920 | 0.2082 | 0.2056 | 0.2946 | 0.751 |
| Ours | 0.3114 | 0.1871 | 0.3860 | 0.4427 | 0.8641 |
| Dataset: HawkI - Real | | | | | |
| DreamBooth LoRA | 0.2929 | 0.1791 | 0.5501 | 0.5088 | 0.9000 |
| Imagic | 0.2960 | 0.1773 | 0.5024 | 0.5048 | 0.9065 |
| Aerial Diffusion | 0.2896 | 0.2084 | 0.1717 | 0.2614 | 0.7606 |
| Ours ($\lambda_{MI} = 1e - 5$) | 0.3077 | 0.1887 | 0.3314 | 0.3956 | 0.8725 |

Table 2: We compare HawkI with various state-of-the-art methods on quantitative metrics indicative of viewpoint and fidelity. Our method achieves the best viewpoint-fidelity trade-off amongst all prior work.

| Method | CLIP | A-CLIP | SSCD | DINO | CLIP-I |
|---|---|---|---|---|---|
| Dataset: HawkI - Real | | | | | |
| Latent Inversion | 0.2940 | 0.1711 | 0.5224 | 0.5148 | 0.9095 |
| w/o "aerial view" | 0.2901 | 0.1641 | 0.6334 | 0.5948 | 0.9495 |
| Ours ($\lambda_{MI} = 1e - 5$) | 0.3077 | 0.1887 | 0.3314 | 0.3956 | 0.8725 |

Table 3: We compare HawkI with latent inversion in the first row, where the diffusion backbone remains frozen, the latent is inverted, and the term 'aerial view' is added to the target prompt. Similarly, when the term 'aerial view' is omitted from the inference text prompt, the model is unable to generate the aerial view of the input image and simply reproduces the input image. Our method achieves the best viewpoint-fidelity trade-off.

| Method | CLIP | A-CLIP | SSCD | DINO | CLIP-I |
|---|---|---|---|---|---|
| Dataset: HawkI - Real | | | | | |
| Latent Inversion | 0.3240 | 0.1711 | 0.5024 | 0.5048 | 0.9193 |
| Ours | 0.3977 | 0.2487 | 0.4114 | 0.4956 | 0.8925 |

Table 4: We compare HawkI with the vanilla baseline corresponding to latent inversion, using the stable diffusion 3.5 backbone. We show significant improvements, demonstrating that our method is effective with different backbone foundational model architectures, and can be used as a plug-and-play method as newer models emerge.

vectors generated from the images1. The SSCD model uses self-supervised contrastive learning with strong differential entropy regularization to create compact descriptor vectors that can be used for efficient and accurate copy detection.

LPIPS Score: The Learned Perceptual Image Patch Similarity (LPIPS) score is a metric used to evaluate the perceptual similarity between two images. It measures how similar two images appear to human observers by comparing features extracted from deep neural networks.

