# OpenReview forum: "HawkI: Homography & Mutual Information Guidance for 3D-free Single Image to Aerial View"
_TMLR — Rejected by TMLR_

### Review · Reviewer_s1tU · 2024-11-26

**Summary Of Contributions:**

This paper introduces a novel approach for generating aerial view images using a single exemplar image and a text description, without relying on 3D information or additional viewpoints. The method leverages a pretrained text-to-image stable-diffusion model, incorporating test-time fine-tuning guided by homography transformations to enable aerial-view synthesis. During inference, mutual information guidance is employed to maintain the semantic integrity of the original image while producing realistic aerial perspectives. Quantitative and qualitative comparisons demonstrate that the proposed method achieves superior fidelity in aerial-view image synthesis.

**Audience:**

Yes

**Claims And Evidence:**

No

**Requested Changes:**

**Questions**
- Could you provide a quantitative and qualitative comparison with a straightforward baseline where the diffusion backbone remains frozen, the latent is inverted, and the term *‘aerial view’* is simply added to the target prompt? Figure 7, second column, appears to illustrate something similar, but it is unclear if the term *‘aerial view’* was included in the target caption. If this approach has already been implemented, I would be interested in seeing the quantitative comparison.
- Could you provide a quantitative and qualitative comparison of your method with and without including the term ‘aerial view’ in the target caption?
- I understand that SD2.1 was used consistently across methods to ensure a fair evaluation. However, could employing a more advanced diffusion UNet backbone (such as SDXL, SSD-1B, or SSD-Vega) enhance the quality of the images generated by your method? While I recognize that this evaluation may be resource-intensive, I think it is important to assess whether your method generalizes well to different backbones and outperforms the simple baseline described above, applied to a scaled backbone.
- How diverse are the generated aerial images? Could you provide additional qualitative examples that demonstrate the variability of the generated images from a single exemplar?
- Why do you use a mix of real and synthetic images for evaluation?
- Can you elaborate the key differences between your text-time optimization for image reconstruction and the optimization process as done in [1] and [2]?
- To what extent would it help to relax the condition of having a single image and instead have a small subset of viewpoints as done in [2]?

**Requested Changes**
- Would it be possible to adjust the figures to make them clearer and easier to interpret?
- Can you provide in the main material or in supplementary a formal definition of the metrics used?
- Section 4: For better readability, could you please insert tables to summarize the quantitative results?
- For better clarity, can you add the term “aerial view” in the captions written in qualitative figures?

[1] Kawar, Bahjat, et al. "Imagic: Text-based real image editing with diffusion models." Proceedings of the IEEE/CVF Conference on Computer Vision and Pattern Recognition. 2023.

[2] Gal, Rinon, et al. "An image is worth one word: Personalizing text-to-image generation using textual inversion." arXiv preprint arXiv:2208.01618 (2022).

**Strengths And Weaknesses:**

**Strengths**
- The method is simple and can be applied to any in-the-wild image.
- Only one single image and one text prompt are required.
- From reported figures, the proposed optimization process seems to be quicker than Imagic’s.
- Using Inverse Perspective Mapping seems to be relevant to guide the diffusion process to generate aerial views.
- The Mutual Information Guidance proves to be effective to preserve the semantics of the original image.

**Weaknesses**
- The paper's presentation could be enhanced for better clarity. The figures are difficult to read and could benefit from improved design and labeling. Additionally, the quantitative results presented in Section 4 would be more comprehensible if organized into tables.
- The aerial point-of-view in the generated images is not particularly striking in the qualitative examples. Relying on a single example image poses a significant constraint for generating aerial perspectives, as while practical, restricts the flexibility  to modify the viewpoint while maintaining image fidelity. Although the use of Mutual Information Guidance effectively preserves semantic consistency, the extent of viewpoint transformation remains limited. For instance, while the quantitative results indicate consistent improvements, the qualitative differences compared to Imagic are less pronounced.
- The visual quality appears to be constrained by the selected stable-diffusion backbone, although this choice can be justified by a fair comparison between methods.
- The evaluation on synthetic images provides limited insights, and the paper would benefit from a more comprehensive ablation study, exploring factors such as different backbones, quantitative evaluation on various viewpoints (back, sides), image diversity, and incorporating additional image views if feasible within the pipeline.
- The paper would benefit from highlighting the key differences in the optimization process compared to [1] and [2].

[1] Kawar, Bahjat, et al. "Imagic: Text-based real image editing with diffusion models." Proceedings of the IEEE/CVF Conference on Computer Vision and Pattern Recognition. 2023.

[2] Gal, Rinon, et al. "An image is worth one word: Personalizing text-to-image generation using textual inversion." arXiv preprint arXiv:2208.01618 (2022).

---

> ### Author Response · Authors · 2024-11-29
>
> Dear Reviewer,
>
> Thanks a lot for your time and valuable feedback to strengthen the paper. We'll work on your suggestions and get back to you at the earliest!
>
> Thanks,
> Authors

---

> > ### Author Response · Authors · 2024-12-24
> > **Response to reviews**
> >
> > Dear Reviewer,
> >
> > Thank you for your valuable feedback. We are thrilled that you find our **method simple and appreciate its applicability to any in-the-wild image**. We are also pleased that you **recognize the use of a single image and text prompt, the speed of optimization compared to Imagic, and the appreciation of our method’s IPM and mutual information-based techniques.**
> >
> > We have addressed your feedback and thoroughly revised the paper to incorporate your comments. Below, we detail the changes we’ve made in response to your suggestions and answer any related questions.
> >
> > **Latent inversion comparison** - yes, the second column in the figure does exactly that. The target caption does include ‘aerial view’. Additionally, we have now added a quantitative comparison in the appendix.
> >
> > **w/o “aerial view” in the caption** - Without ‘aerial view’ in the caption, there is **no change in viewpoint**, the method simply reproduces the input image. We have added a few qualitative results and the quantitative results to the appendix.
> >
> > **Another backbone** - we apply our method to SD-3.5 (as it is more advanced than SDXL and SSD Vega). We show the comparison with the simple latent inversion baseline as suggested by the reviewer and report the results in the appendix.
> >
> > **Diversity of images** - we have added it to the appendix.
> >
> > **Why synthetic as well as real images?** We utilize synthetic images to demonstrate the effectiveness of our method across a diverse set of input images, by synthetically generating varying complexities. Additionally, we use real images from Unsplash to showcase the applicability of our method in complex, in-the-wild scenes.
> >
> > **Difference between our method and [1] and [2]** - In [1] (Imagic), the target caption is used to optimize the text embedding for the input image, followed by fine-tuning the UNet. Since the change desired in [1] is **local**, inferencing at the target caption now provides the desired output. In our case, the desired change is global. We use the caption of the input image (without the 'aerial view' text) to optimize the text embedding, followed by optimizing the UNet for the image. The purpose of optimizing the text embedding is to increase variance and prevent overfitting to the input image, which is necessary for generating the target image, while optimizing the UNet helps learn the characteristics of the input image.
> >
> > In [2] (Textual Inversion), a new text token is assigned to the given concept and learned to reconstruct the concept. There is no new text token in our case.
> >
> > **Multiple images per scene** - The model would then have more information about the scene, enhancing the fidelity of the generated target image relative to the input image. This would also increase the model's variance, enabling better viewpoint generation. However, it is **impractical to obtain multiple views of in-the-wild input scenes**; thus, using a single image remains the pragmatic approach.
> >
> > **Adjusting figures** - Yes, we have formatted them to occupy the entire page width. Please let us know if any readability issues persist, and we would be happy to make further adjustments.
> >
> > **Metrics definition** - We have now added them to the appendix.
> >
> > **Tables for Sect 4** - We have now added them to the appendix.
> >
> > **`Aerial view’ in the figures** - We chose not to include 'aerial view' in the figure because the training process uses only the text, while the **inference process uses 'aerial view' + text**. We wanted to avoid confusing readers with this distinction. We have clarified this inference text prompt in the method section.
> >
> > We hope that the new version of our paper and the accompanying rebuttal comment address all your questions. We have diligently incorporated your feedback, which has significantly strengthened our paper.
> >
> > We sincerely thank you for your time, assistance, and valuable comments. Should there be any further changes you would like us to make, please let us know, and we would be happy to incorporate them.
> >
> > **Wishing you a Merry Christmas and a Happy New Year!**
> >
> > Best regards,
> > Authors

---

### Review · Reviewer_yRBy · 2024-12-16

**Summary Of Contributions:**

This paper studies text-guided aerial-view image generation without multi-view or 3D information. The input is a reference image from another view, along with a text prompt. The proposed method is based on a T2I model, with the proposed IPM method to achieve mutual info guidance, to accomplish this task with a test-time optimization.

**Audience:**

Yes

**Claims And Evidence:**

Yes

**Requested Changes:**

- Is the proposed method very sensitive to the hyperparameters? The paper mentioned in "Training details" that there are many hyperparameters, and it seems there are a lot of hypermeters that need to be selected. Is there a rule for selecting the hyperparameters?
- The method seems compatible with DreamBooth, that one way is to first train DreamBooth to memorize the appearance of the object, and then apply the proposed method to generate the aerial view. Will this achieve an improvement?
- Will the method be able to generate the aerial view for more detailed and fine-grained content like human faces and poses?

**Strengths And Weaknesses:**

### Pros
- The idea of using IPM is novel and tailored for aerial view generation.
- Various baselines are compared in the experiments, along with an extensive ablation study.
- The quality of the generated method is high, which outperforms the baselines. The viewpoint is lifted to an aerial view, and the appearance of the main object in the original image is also preserved well.

### Cons
- Test-time optimization should be regarded as a shortcoming. It is like SDS (score distillation sampling) that requires a per-task optimization, that is unstable, time-consuming, and/or even requires per-task hyperparameter tuning.
- According to "Training details", it seems that each editing task needs to choose lots of hyperparameters.
- In Fig.6, the generation results of other views (other than the aerial view) seems not reasonable. The side, bottom, or the back view, at the most of the time, has only a minor difference compared with the input image, and the preservation of other contents (e.g., floor) are not good.

---

> ### Author Response · Authors · 2024-12-16
>
> Dear Reviewer,
>
> Thanks a lot for your time and valuable feedback to strengthen the paper. We'll work on your suggestions and get back to you at the earliest!
>
> Thanks, Authors

---

> ### Author Response · Authors · 2024-12-24
> **Response to reviews**
>
> Dear Reviewer,
>
> Thank you for your valuable feedback. We are elated that you find our **method novel, appreciate the comparisons and ablation study** and also **appreciate the quality of the generated images in generating aerial viewpoint while preserving the characteristics of the input images.**
>
> We have addressed your feedback and have thoroughly revised the paper to incorporate your comments. Below, we explain the changes we’ve made, in accordance with your comments, and also answer pertaining questions.
>
> **Test-time optimization** - Indeed, test-time optimization poses a limitation in terms of time consumption and stability, and addressing these limitations will be a focus of future work. Our proposed method is not sensitive to hyperparameters per image. In fact, we **do not finetune any hyperparameters for individual images**. We used a small validation subset of images to set the hyperparameters and have since used the same values for all images without any changes.
>
> To select hyperparameters, we used the same parameters as Aerial Diffusion for optimizing the text embedding and fine-tuning the diffusion UNet on the input image. For the homography image, we halved the number of iterations while retaining all other parameters. The only hyperparameter requiring consideration was mutual information, which we tuned using the validation subset. Thus, there is no per-task or per-image hyperparameter tuning involved. We have added these details to the appendix.
>
> **DreamBooth type model comparisons** - Yes, that is absolutely true. In fact, in our DreamBooth comparisons, we do exactly that. We find that while DreamBooth can achieve minor changes, such as transforming a log cabin into a glass cabin, it struggles with executing larger transformations, like shifting from a front to an aerial viewpoint (please refer to the newly added example in the appendix). **The superiority of our method over a DreamBooth-style approach** is evidenced by our qualitative and quantitative results.
>
> **Human faces and poses** - Yes, for generating human faces and poses, we need a more advanced foundational model backbone that comprehends general pose and face generation. This would enable us to customize it for view transformations of the provided in-the-wild input images containing faces or poses. Our method is a **plug-and-play solution and can be integrated with other foundational model backbones for human faces and poses.**
>
> **Generation of other views** - Yes, the experiments we have presented are preliminary and demonstrate the potential of our method for generating alternative viewpoints. These experiments lay the groundwork for further exploration and development in the field of generic novel view synthesis.
>
> We hope that the new version of our paper and the accompanying rebuttal comment address all your questions. We have diligently incorporated your feedback, which has significantly strengthened our paper.
>
> We sincerely thank you for your time, assistance, and valuable comments. Should there be any further changes you would like us to make, please let us know, and we would be happy to incorporate them.
>
> **Wishing you a Merry Christmas and a Happy New Year!**
>
> Best regards,
> Authors

---

### Review · Reviewer_cBZx · 2024-12-16

**Summary Of Contributions:**

This work introduces "Hawkl", a method to synthesize aerial-view images from a reference image alongside its text description. This approach involves optimizing CLIP text embeddings and diffusion UNet LoRA layers alternately in test-time optimization, applied to the original image and its modified version by Inverse Perspective Mapping (IPM). Additionally, it incorporates mutual information-based guidance akin to classifier guidance during inference sampling, enhancing content consistency between the generated and reference images.

**Audience:**

Yes

**Claims And Evidence:**

No

**Requested Changes:**

Please refer to the "Weaknesses" section for more details.

**Strengths And Weaknesses:**

# Strengths

1. The task setting that generates aerial views of any "in-the-wild" images is intriguing and meaningful
2. Qualitative and quantitative experiments are thorough. Ablation studies are well-conducted to evaluate the effectiveness of some design choices

---

# Weaknesses

1. **Motivation for this work is unconvincing**. The manuscript repeatedly highlights that a key advantage of the proposed method is its ability to function *without any multi-view or 3D information*. However, the rationale for excluding multi-view or 3D information is insufficiently discussed. For example, what is the essential bottleneck of the aerial-view synthesis task? If the issue lies in the lack of multi-view or 3D datasets, this claim does not hold, as numerous large-scale multi-view/3D datasets have emerged, such as Objaverse(-XL) [1, 2], MVImageNet (V1 and V2) [3, 4], DL3DV [5], etc., and numerous video datasets that could serve as potential data sources. Furthermore, as discussed in **Sec. 4.8**, many multi-view or 3D pretrained models incorporating 3D priors could be beneficial for this task. This lack of justification could represent a fundamental flaw of the proposed approach.

2. **Proposed techniques do not seem to make sense**. The manuscript asserts that utilizing *mutual information guidance* during sampling preserves semantic consistency between generated and reference images. However, the *used mutual information appears to be computed at the RGB pixel statistical level rather than the image or semantic distribution level*. This is evident as the authors construct 2D histograms to calculate marginal and joint PDFs, whereas the scores provided by diffusion models should ideally be employed for mutual information computation, like Distribution Matching Distillation (DMD) [6, 7]. Consequently, the so-called "mutual information guidance" seems to primarily focus on aligning color distributions between generated and reference images. Furthermore, the use of inverse perspective-mapped images as pseudo-supervision also seems to be strange, as these images significantly deviate from the expected aerial views and are heavily distorted compared to natural image distributions.

3. **Generation quality of the proposed method is poor and some comparisons are unfair**. As previously mentioned, due to some design flaws, the generated images are over-saturated and deviate significantly from the reference images, although the authors compared their method to some baseline techniques, which are considered to be out of date. Meanwhile, methods proposed for object novel view synthesis (Zero123 and Zero123++) are compared with the proposed method, while scene novel view synthesis methods, such as GeNVS [8] and ZeroNVS [9], are not discussed.

4. **Presentation of this work requires improvement**. For instance, the citation format is incorrectly applied as `\cite` instead of `\citep`. Additionally, the term "CLIP text-image embedding" is unclear and likely intended to mean "CLIP text embedding", as "CLIP text-image embedding" typically refers to a pair of embeddings that include both text and image components. Some expressions also lack clarity, such as "bias-variance trade-off", which is commonly used in the context of deterministic models. However, in the context of generative models, "bias" and "variance" are not well-defined, though I could understand these terms might correspond to "fidelity" and "diversity" respectively.

---

[1] Objaverse: A universe of annotated 3d objects. CVPR 2023.

[2] Objaverse-XL: A universe of 10m+ 3d objects. NeurIPS 2023.

[3] Mvimgnet: A large-scale dataset of multi-view images. CVPR 2023.

[4] MVImgNet2.0: A Larger-scale Dataset of Multi-view Images. TOG 2024.

[5] DL3DV-10K: A large-scale scene dataset for deep learning-based 3d vision. CVPR 2024.

[6] One-step diffusion with distribution matching distillation. CVPR 2024.

[7] Improved Distribution Matching Distillation for Fast Image Synthesis. NeurIPS 2024.

[8] Generative novel view synthesis with 3d-aware diffusion models. ICCV 2023.

[9] Zeronvs: Zero-shot 360-degree view synthesis from a single real image. CVPR 2024.

---

> ### Author Response · Authors · 2024-12-16
>
> Dear Reviewer,
>
> Thanks a lot for your time and valuable feedback to strengthen the paper. We'll work on your suggestions and get back to you at the earliest!
>
> Thanks, Authors

---

> ### Author Response · Authors · 2024-12-24
> **Response to reviews (1/2)**
>
> Dear reviewer,
>
> Thank you for your valuable feedback.
>
> We are delighted that you find our task **insightful and meaningful**. We appreciate your recognition of our **comprehensive qualitative and quantitative experiments**, as well as the ablation studies.
>
> We have addressed your feedback and made thorough revisions to the paper to incorporate your comments. Below, we outline the changes made in response to your suggestions and answer related questions.
>
> **Motivation for 3D-free**: Collecting and training on 3D data at a large scale is **expensive and unsustainable**. Even with 3D datasets like Objaverse or MVImageNet, the data is specific to certain scenes and objects. While training on such data can yield good performance for those specific scenarios, it offers **limited generalization** to other in-the-wild or out-of-distribution scenes. This limitation is due to the relatively small size of these datasets (around 800k objects), and scaling to the level of image datasets like LAION-5B for 3D objects is prohibitively expensive and challenging.
>
> On the other hand, front-view 2D images are more readily available, and there are 2D text-image diffusion models trained on billions of data points. Hence, it is advantageous to tackle fundamental problems in a data-efficient (or 3D-free) manner using only 2D information. In this paper, our goal was to **push the boundaries of 3D-free aerial-view generation from a single image, achieving results comparable to methods that use stable diffusion with 800k 3D objects.**
>
> As a subsequent objective, we explored utilizing pretrained 3D models. While 3D models provide enhanced 3D controllability, their generalizability is limited due to data scale issues. Thus, we posed the question: Can HawkI be combined with 3D approaches to leverage the benefits of both 3D and 3D-free methods? **We found that this combination harnesses the data efficiency and generalizability of HawkI while generating aerial views of any in-the-wild image with the 3D control benefits of pretrained 3D models from existing Objaverse datasets, which may not be broadly generalizable. This paves the way for new research directions.** We’ve also added this explanation to the appendix.
>
> **Proposed techniques do not seem to make sense**: Mutual information is **indeed computed in the latent space**, as rightly suggested by the reviewer. It is not computed at the RGB pixel statistical level.
>
> Regarding the use of Inverse Perspective Mapping, our goal was to address the problem in a 3D-free manner. For supervision, we used classical computer vision techniques for weak or pseudo guidance. Although distorted, this approach provides sufficient information for our powerful HawkI model to extract and achieve the desired goal. As a future research step, priors from pretrained 3D models (without additional 3D data) can be used for **weak guidance**, as shown in section 4.8.
>
> Our design choices are **validated** by our extensive experimentation, analysis, and ablation studies.
>
>
> **Comparisons and generation quality**
>
> **Generation Quality:** We acknowledge the feedback regarding the generation quality. However, we would like to emphasize that our method shows **remarkable advancements over previous work**, both qualitatively and quantitatively. While it may not be a 100% solution, it certainly paves the way for future improvements, as is the nature of ongoing research.
>
> **Comparisons:**
>
> **Image Manipulation Approaches:** Since our method utilizes a text-based image manipulation approach, we compare it with techniques like DreamBooth and Imagic.
>
> **Single Image Stable Diffusion-Based 3D Model Comparisons:** Our 3D-free method employs a stable diffusion backbone, so we compare it with methods that use stable diffusion combined with 3D objects for training. Both approaches take a single image as input to generate aerial views.
>
> **Stable Diffusion Variations:** We also compare our method with others such as ControlNet.
>
> **Text-Based Single Image to Aerial View Approaches:** Additionally, we compare our method with other 3D-free approaches like Aerial Diffusion, which use stable diffusion for generating aerial views from single images.
>
> **GeNVS and ZeroNVS:** Methods such as GeNVS and ZeroNVS **utilize 3D priors** in the form of 3D feature fields or distill 3D information from NeRFs for specific scenes. However, this **dependence on 3D data restricts their applicability** to arbitrary, out-of-distribution, in-the-wild scenes, which is the central focus of our paper.
>
> In contrast, our method is entirely 3D-free. While Zero123++ and Zero123 require 800K 3D data for training, our approach uses pretrained 3D models without the need for additional 3D data during inference.

---

> ### Author Response · Authors · 2024-12-24
> **Response to reviews (2/2)**
>
> **Presentation issues**: We have fixed all issues related to cite format, CLIP embedding and bias-variance.
>
> We hope that the new version of our paper and the accompanying rebuttal comment address all your questions. We have diligently incorporated your feedback, which has significantly strengthened our paper.
>
> We sincerely thank you for your time, assistance, and valuable comments. Should there be any further changes you would like us to make, please let us know, and we would be happy to incorporate them.
>
> **Wishing you a Merry Christmas and a Happy New Year!**
>
> Best regards,
> Authors

---

### Author Response · Authors · 2024-12-24
**Rebuttal and revised pdf**

Dear Reviewers and Area Chair,

We would like to express our heartfelt gratitude for your time and efforts in reviewing our paper. We have diligently addressed your feedback and made the necessary changes. Additionally, we have answered your questions related to our paper. We sincerely hope that the revised version of our paper, along with our rebuttal, satisfies your concerns.

We look forward to further engaging discussions and are happy to make any further changes as required.

Thank you once again for your valuable feedback and support.

Best regards,
Authors

---

### Author Response · Authors · 2025-01-09
**Following up**

Dear Reviewers and Area Chair,

We hope you are doing well! Wishing you a happy new year!

We hope that the revised paper has addressed your feedback. Please let us know if there are any further questions that we can answer and if there are any further changes that you would like to see in the paper.

Thanks,
Authors

---

### Decision · Action_Editor_qXef · 2025-01-31

**Recommendation:** Reject

**Comment:**

The reviewers expressed many concerns in their initial reviewer, encompassing the following aspects:
- Experiments (qualitative results, comparisons to other methods)
- Presentation (clarity, quality)
- Motivation of the work
- Some methodological choices

In their final recommendation, Reviewers yRBy and s1tU acknowledged that the authors' responses addressed some of their concerns. Nevertheless, Reviewers cBZx and s1tU were not fully convinced by these responses and recommended rejection, with Reviewer s1tU specifically stating not being convinced by the evaluation and ablation studies, as also mentioned in the Claims and Evidence box above.

**Audience:**

Yes.

**Claims And Evidence:**

The claims are not fully supported by clear evidence. In particular, all three reviewers raised concerns about some aspects of the experiments. While the authors addressed some of these concerns, they fail to fully convince Reviewers cBZx and s1tU, with Reviewer s1tU stating in their final recommendation that "the evaluation and ablation is not comprehensive enough and qualitative results do not particularly illustrate the aerial point of view."